# RETHINKING THE SYMMETRY-PRESERVING CIRCUITS FOR CONSTRAINED VARIATIONAL QUANTUM ALGORITHMS

**Ge Yan, Hongxu Chen, Kaiseng Pan, Junchi Yan**[*]
Department of Computer Science and Engineering, Shanghai Jiao Tong University
{yange98,mechta_chx,pks0813,yanjunchi}@sjtu.edu.cn

## ABSTRACT

With the arrival of the Noisy Intermediate-Scale Quantum (NISQ) era, Variational Quantum Algorithms (VQAs) have emerged as a popular paradigm to obtain possible quantum advantage in the relatively near term. In particular, how to effectively incorporate the common symmetries in physical systems as hard constraints in VQAs remains a critical and open question. In this paper, we revisit the Hamming Weight (HW) preserving ansatz and establish the link from ansatz to various symmetries and constraints, which both enlarges the usage of HW preserving ansatz and provides a potentially coherent solution for constrained VQAs. Meanwhile, we utilize the quantum optimal control theory and quantum overparameterization theory to analyze the capability and expressivity of HW preserving ansatz and verify these theoretical results on the unitary approximation problem. We conduct detailed numerical experiments on two well-studied symmetry-preserving problems, namely ground state energy estimation and feature selection in machine learning. The superior performance demonstrates the efficiency and supremacy of the proposed HW preserving ansatz on constrained VQAs.

## 1 INTRODUCTION

Over the decade, Variational Quantum Algorithms (VQAs) (Cerezo et al., 2021; Tilly et al., 2022) have received increasing attention and numerous studies have been conducted to seek potential quantum supremacy. With the arrival of the Noisy Intermediate-Scale Quantum (NISQ) era (Preskill, 2018; Bharti et al., 2022), the pace of exploring new VQAs has been further accelerated, as VQAs have shown the potential to obtain quantum advantage in the foreseeable future on NISQ devices (Cerezo et al., 2021). However, typical VQAs such as Quantum Approximate Optimization Algorithm (QAOA) (Farhi et al., 2014) for Quadratic Unconstrained Binary Optimization (QUBO), and UCCSD (Romero et al., 2018) for ground state energy estimation are not natively designed to deal with (hard) constraints. It remains an open problem for addressing and incorporating the symmetries and constraints in the VQAs.

We propose to resort to Hamming Weight (HW) preserving ansatz (Kerenidis et al., 2021) to introduce hard constraints to the VQAs instead of modeling the constraints as a regularizer in the Hamiltonian as widely done in literature (Chieza et al., 2020; Khumalo et al., 2021). The HW preserving ansatz operate in a restricted $\binom{n}{k}$-dimensional subspace with parameterized HW preserving gates. We argue and show that HW preserving ansatz can deal with various symmetries in the physical systems and it could be used as a sound alternative for the Hardware Efficient Ansatz (HEA) for constrained VQAs. We further utilize the quantum optimal control (Schirmer et al., 2001) and quantum overparameterization (Larocca et al., 2023) theories to lay a rigorous theoretical foundation for the HW preserving ansatz. Specifically, quantum optimal control theory can help us decide whether a HW preserving gate with certain connectivity is universal. Analysis of the trainability ensures that we can evolve the quantum state without the presence of barren plateaus. While the overparameterization theory provides us with the guide that how many parameters (layers) one needs to obtain feasible results.

---

[*]Correspondence author. This work was in part supported by NSFC (92370201, 62222607).

We provide experimental results on two constrained VQA problems, namely ground state energy estimation (Kandala et al., 2017) and feature selection (Chandrashekar & Sahin, 2014) in machine learning. We use two different HW preserving gates with two different connectivity to illustrate the capability and expressivity of different HW preserving ansatz. Specifically, one of the gates is Reconfigurable Beam Splitter (RBS) gate as proposed in literature (Cherrat et al., 2023), and the other BS gate is generated through theoretical analysis and then is verified by empirical results. Both HW preserving gates demonstrate great efficiency on constrained VQA problems. BS gate is universal on the $\binom{n}{k}$-dimensional subspace and is able to solve the unitary approximation problem on the subspace. However, RBS gate shows a better convergence rate on easier problem such as the ground state energy estimation but only with full connectivity. **Our contributions are:**

1) We revisit the Hamming weight (HW) preserving ansatz and link the ansatz to popular symmetries and constraints in physical systems, thereby expanding the utility of the HW preserving ansatz beyond their currently limited applications (Landman et al., 2022; Cherrat et al., 2023).

2) To our best knowledge, we are the few (perhaps the first) to provide a theory for the capability, expressivity, and trainability of HW preserving ansatz. We are able to generate a new gate based on our theories and numerical results on the task of unitary approximation verify our hypotheses.

3) We conduct detailed numerical experiments on symmetry-preserving ground state energy estimation and feature selection in machine learning, which are two popular tasks. The superior performance demonstrates the efficiency of the HW preserving ansatz on constrained VQAs.

## 2 PRELIMINARIES

In this section, we will introduce the definition of the HW preserving quantum circuit and why it is useful in customizing the symmetries and other constraints in the VQE problems. We will further introduce several gates that satisfy the constraints of HW preserving.

### 2.1 HAMMING WEIGHT PRESERVING QUANTUM CIRCUIT

We have discussed several types of symmetries at the physics level and we now consider these symmetries and constraints on the quantum circuits. These symmetries and constraints as discussed in the related work all ensure that the number of 1s in the quantum state does not change. Recall the definition of Hamming distance and Hamming weight (Def. B.1 and Def. B.2), we find that these states in the constrained subspace all have the same Hamming weight. Thus we can employ a Hamming weight preserving circuit to solve the aforementioned problems in a limited subspace.

To make sure the quantum circuit will not generate any quantum state that have different Hamming weight, we need to find proper gates that can preserve the Hamming weight of the initial states. The most common gate we can find is the SWAP gate and the RZZ gate with the unitary matrices

$$\mathbf{U}_{SWAP} = \begin{pmatrix} 1 & 0 & 0 & 0 \\ 0 & 0 & 1 & 0 \\ 0 & 1 & 0 & 0 \\ 0 & 0 & 0 & 1 \end{pmatrix}, \quad \mathbf{U}_{RZZ} = e^{-i\frac{\theta}{2}} \begin{pmatrix} 1 & 0 & 0 & 0 \\ 0 & e^{i\theta} & 0 & 0 \\ 0 & 0 & e^{i\theta} & 0 \\ 0 & 0 & 0 & 1 \end{pmatrix}. \quad (1)$$

These two gates ensure that they do not change the amplitudes of state $|00\rangle$ and $|11\rangle$. However, it is still not enough to build a trainable circuit for HW preserving problems. Cherrat et al. (2023) utilized another gate called the Reconfigurable Beam Splitter (RBS) gate as the basic element, and (Hadfield et al., 2019) proposed XY-mixer to construct constrained QAOA. RBS gate and XY-mixer are more trainable compared to RZZ and SWAP gate with a possible decomposition in Fig. 5.

$$\mathbf{H}_{RBS} = \begin{pmatrix} 0 & 0 & 0 & 0 \\ 0 & 0 & -i & 0 \\ 0 & i & 0 & 0 \\ 0 & 0 & 0 & 0 \end{pmatrix}, \mathbf{U}_{RBS}(\theta) = e^{i\theta \mathbf{H}_{RBS}} = \begin{pmatrix} 1 & 0 & 0 & 0 \\ 0 & \cos(\theta) & \sin(\theta) & 0 \\ 0 & -\sin(\theta) & \cos(\theta) & 0 \\ 0 & 0 & 0 & 1 \end{pmatrix}, \quad (2)$$

$$\mathbf{H}_{XY} = \begin{pmatrix} 0 & 0 & 0 & 0 \\ 0 & 0 & 1 & 0 \\ 0 & 1 & 0 & 0 \\ 0 & 0 & 0 & 0 \end{pmatrix}, \quad \mathbf{U}_{XY}(\theta) = e^{i\theta \mathbf{H}_{XY}} = \begin{pmatrix} 1 & 0 & 0 & 0 \\ 0 & \cos(\theta) & -i\sin(\theta) & 0 \\ 0 & -i\sin(\theta) & \cos(\theta) & 0 \\ 0 & 0 & 0 & 1 \end{pmatrix}. \quad (3)$$

Consider a $n$-qubit quantum circuit with HW $k$, we have $d_k = \binom{n}{k}$ basis states in the HW subspace. The SWAP, RZZ and RBS gates all preserve the subspace with HW as $k$. The main character of these 2-qubit gates is that they only act on two basis states $|01\rangle$ and $|10\rangle$. Following this condition, we can theoretically construct our own gate that is HW preserving. Note that the dimension of the HW preserving circuit is $\binom{n}{k}$, which is smaller than the dimension of the whole Hilbert space as $2^n$ (especially when $k$ is relative small compared to $n$). This suggests some interesting properties.

## 2.2 DYNAMICAL LIE ALGEBRA

To understand the expressivity and capability of the HW preserving ansatz, we need to utilize a mathematics tool, which is Dynamical Lie Algebra (DLA) (Zeier & Schulte-Herbrüggen, 2011; d'Alessandro, 2021). Since quantum circuits are unitary transformations, we can use Lie algebra and Lie group to analyze the properties of them. However, quantum machine learning introduces parameters into the unitary transformations, so we need to further extend the Lie algebra to dynamical Lie algebra. The unitary matrix $\mathbf{U}(\boldsymbol{\theta})$ of a QML circuit with $L$ layers is of the form:

$$\mathbf{U}(\boldsymbol{\theta}) = \prod_{l=1}^{L} \mathbf{U}_l(\boldsymbol{\theta}_l) = \prod_{l=1}^{L} \prod_{p=1}^{P} e^{i\theta_{lp}\mathbf{H}_p} \tag{4}$$

where $l$ indicates the layer, $\mathbf{H}_p$ are the Hermitian matrices that generate the unitary matrix $\mathbf{U}_l(\boldsymbol{\theta}_l)$. $\boldsymbol{\theta}_l = \{\theta_{l1}, \theta_{l2}, \cdots, \theta_{lp}\}$ is the parameters in the $l$-th layer and $\boldsymbol{\theta} = \{\boldsymbol{\theta}_1, \boldsymbol{\theta}_2, \cdots, \boldsymbol{\theta}_L\}$ is the parameters in the whole circuit. The generator of each layer in the ansatz can be defined as follows:

**Definition 2.1** *The set of generators: Consider a single layer of parameterized quantum circuit $\mathbf{U}_l(\boldsymbol{\theta})$ in Eq. 4. We define a group of Hermitian matrices that generate the unitary matrices $\mathbf{U}(\boldsymbol{\theta})$ as a set of generators $\mathcal{G} = \{\mathbf{H}_p\}_{p=1}^{P}$, where $|\mathcal{G}| = P$.*

With the set of generators by Definition 2.1, we introduce a well-established concept:

**Definition 2.2** *Dynamical Lie Algebra (DLA): Consider the set of generators defined in Definition 2.1, the DLA $\mathfrak{g}$ is defined as:*

$$\mathfrak{g} = span\langle i\mathbf{H}_1, i\mathbf{H}_2, \cdots, i\mathbf{H}_P\rangle_{Lie}, \tag{5}$$

*where $\langle \cdot \rangle_{Lie}$ denotes the Lie closure. The set of generators is obtained by repeated take the commutators of the elements in the set.*

We denote the corresponding Dynamical Lie group of DLA $\mathfrak{g}$ as $\mathbb{G}$. The reachable unitary matrices from arbitrary parameters $\boldsymbol{\theta}$ have the following properties (Larocca et al., 2023):

$$\{\mathbf{U}(\boldsymbol{\theta})\}_{\boldsymbol{\theta}} \subseteq \mathbb{G} \subseteq \mathcal{SU}(N), \tag{6}$$

where $\mathcal{SU}(N)$ denotes the special unitary group and $d$ denotes the dimension.

## 3 INTERPRETABILITY OF THE HW PRESERVING ANSATZ

In this section, we will discuss the capability and expressivity of the proposed HW preserving ansatz. We also introduce definitions and theories for dynamical lie algebra, quantum control system and quantum overparameterization. With these knowledge, we are able to determine if a gate is universal in the $d_k$-dimensional subspace, and estimate the number of parameters (circuit depth) needed in the circuits. We conduct experiments on the unitary approximation problem to verify these theories.

### 3.1 QUANTUM OPTIMAL CONTROL ON HW PRESERVING SPACE

Now we introduce the quantum control system and how to get complete controllability over a given quantum system. Consider a $N$-dimensional quantum system, the Hamiltonian of which can be denoted as $\hat{\mathbf{H}}$. If we denote the initial state as $|\psi_0\rangle$, the time-independent Schrödinger equation is

$$\hat{\mathbf{H}}\mathbf{U}(\boldsymbol{\theta})|\psi_0\rangle = E\mathbf{U}(\boldsymbol{\theta})|\psi_0\rangle, \tag{7}$$

where $E$ is a scalar and stands for the energy. Notice that not all the states in the space can be reached by the dynamical unitary matrix, unless the dynamical Lie group generated by $\mathcal{G}$ is equal to the special unitary group $\mathcal{SU}(N)$.

**Lemma 3.1** *A quantum system $\hat{\mathbf{H}}$ is completely controllable if $\{\mathbf{U}(\boldsymbol{\theta})\}_{\boldsymbol{\theta}} = \mathbb{G} = \mathcal{SU}(N)$.*

Complete controllability indicates that any target state in the space can be dynamically reached from any initial state with a proper unitary matrix. We can further conclude that complete controllability implies that any unitary matrices in the space can be approximated to an arbitrary precision by $\mathbf{U}(\boldsymbol{\theta})$. Ramakrishna et al. (1995) has shown that if the dimension of DLA $\mathfrak{g}$ generated by the operators $\mathcal{G}$ is $N^2$ then the dynamical Lie group of the system is $\mathcal{SU}(N)$. Therefore, we have

**Theorem 3.2** *(Ramakrishna et al., 1995) A necessary and sufficient condition for complete controllability of a quantum system $\hat{\mathbf{H}}$ is that the dimension of the DLA $\mathfrak{g}$ is $N^2$.*

This theorem gives a simple way to verify the controllability of a quantum system, namely by computing the dimension of the Lie algebra generated by the set of generators $\mathcal{G}$. The detailed computational method to calculate the dimension of DLA is described in Alg. 1 (Schirmer et al., 2001).

Notice that different connectivity of the physical qubits can lead to different dimensions in the DLA since they will have a different set of generators. Take RBS gate as an example, if all the qubits are fully connected (we can apply 2-qubit gates to any pair of qubits), we will have $\frac{1}{2}n(n-1)$ generators in $\mathcal{G}$ and the corresponding dimension of DLA is $\frac{1}{2}d_k(d_k - 1)$. However, if all the physical qubits are linked as a circle (each qubit can only be connected with its two neighbors), we will have $n$ generators in $\mathcal{G}$ and the dimension of DLA is $\frac{1}{2}n(n - 1)$ (illustrated in Fig. 1(b)).

## 3.2 OVERPARAMETERIZATION FOR HW PRESERVING ANSATZ

Once we know which ansatz can have full controllability of the system, it is critical to know the number of parameters or layers we need to build such ansatz. Larocca et al. (2023) provides a theory to link the overparameterization phenomenon to the dimension of DLA. Overparameterization has certain implications for the trainability of an ansatz. Underparameterized ansatz will lead to local minima and overparameterized ansatz can have better properties and lead to a better landscape for optimization. We first define a crucial concept in the overparameterization theory.

**Definition 3.3** *Quantum Fisher Information Matrix (QFIM): Consider quantum state $|\psi_\mu\rangle$ and a set of parameters $\boldsymbol{\theta} = \{\theta_1, \cdots, \theta_M\}$. The QFIM is defined as an $M \times M$ matrix:*

$$[F_\mu(\boldsymbol{\theta})]_{ij} = 4\,\mathrm{Re}\left( \langle \partial_i \psi_\mu(\boldsymbol{\theta})|\partial_j \psi_\mu(\boldsymbol{\theta})\rangle - \langle \partial_i \psi_\mu(\boldsymbol{\theta})|\partial \psi_\mu(\boldsymbol{\theta})\rangle \langle \partial \psi_\mu(\boldsymbol{\theta})|\partial_j \psi_\mu(\boldsymbol{\theta})\rangle \right), \quad (8)$$

*where $M$ is the number of parameters, $|\partial_i \psi_\mu(\boldsymbol{\theta})\rangle = \partial |\psi_\mu(\boldsymbol{\theta})\rangle / \partial \theta_i = \partial_i |\psi_\mu(\boldsymbol{\theta})\rangle$, $\theta_i \in \boldsymbol{\theta}$.*

The overparameterization phenomenon can then be linked to the rank of QFIM.

**Definition 3.4** *Overparameterized QNN: A QNN is defined overparameterized if increasing the number of parameters $M$ past a minimal bound $M_c$ does not further increase the rank of any QFIM:*

$$\max_{M \geq M_c, \boldsymbol{\theta}} rank[F_\mu(\boldsymbol{\theta})] = R_\mu \quad (9)$$

Larocca et al. (2023) further shows that for the general case of QNN, we have:

$$M_c \sim dim(\mathfrak{g}), \quad (10)$$

where $M_c$ is the bound of the number of parameters to reach overparameterization, and $\mathfrak{g}$ is the DLA correspond to the ansatz. When the number of parameters is larger than $M_c$, the ansatz will reach its maximum capacity and further adding parameters will not improve the performance. The theory of overparameterization is important for us to decide the number of gates or layers we need to solve certain problems using a given HW preserving ansatz.

## 3.3 GENERATING UNIVERSAL HW PRESERVING GATES

Now we have rigorous theories for HW preserving gates as well as HW preserving ansatze. We can use the quantum control system theory to generate a gate and decide the connectivity of physical qubits we need to have full controllability and then use the overparameterization theory to decide the number of gates (layers) we need in the ansatz (see Fig. 1 for details).

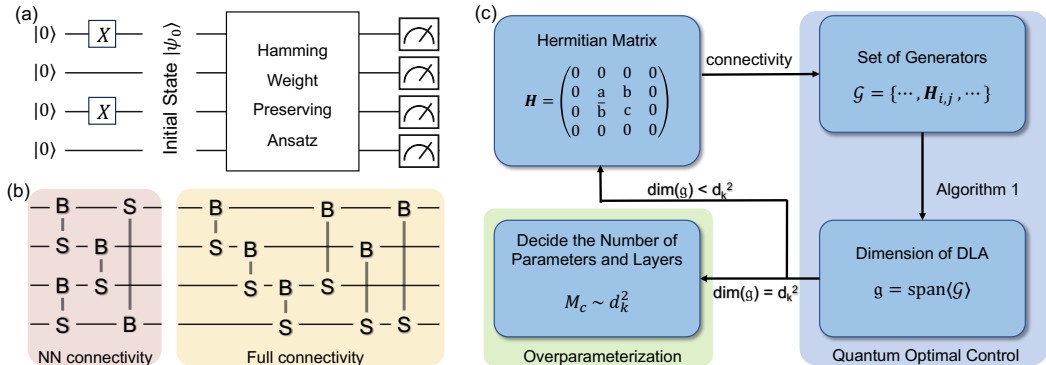

Figure 1: **Pipeline to generate universal HW preserving gates.** (a) The overall circuit of the HW preserving ansatz. We require $k$ pauli-x gates to generate an initial state and then apply the HW preserving ansatz. (b) Examples of two different connectivity. (c) Pipeline to generate universal HW preserving gates and ansatz. We utilize the quantum optimal control and quantum overparameterization theory to help design the gate as well as the ansatz.

Following a similar pattern as SWAP, RZZ and RBS gate, we can generate a two-qubit HW preserving gate by only operating on states $|01\rangle$ and $|10\rangle$. The Hermitian matrix can be of the form:

$$
\mathbf{H}_{HW} = \begin{pmatrix} 0 & 0 & 0 & 0 \\ 0 & a & b & 0 \\ 0 & \bar{b} & c & 0 \\ 0 & 0 & 0 & 0 \end{pmatrix},
\tag{11}
$$

where $a, c \in \mathbb{R}$, $b \in \mathbb{C}$, and $\bar{b}$ denotes the complex conjugate of $b$ since $\mathbf{H} = \mathbf{H}^\dagger$. We only have three variables in the matrix which is simple for us to create new gates. We can then convert the Hermitian matrix to a Unitary matrix that can be further decomposed and applied on a quantum circuit. As an embodiment without loss of generality, we give the following example of Hamiltonian for a HW preserving gate as BS. We start from $\mathbf{H}_{RBS}$ and then repeatedly alter $a, b$ and $c$ to make it universal:

$$
\mathbf{H}_{BS} = \begin{pmatrix} 0 & 0 & 0 & 0 \\ 0 & \frac{1}{2} & \frac{1+i}{2\sqrt{2}} & 0 \\ 0 & \frac{1-i}{2\sqrt{2}} & \frac{1}{2} & 0 \\ 0 & 0 & 0 & 0 \end{pmatrix}, \quad \mathbf{U}_{BS}(\theta) = \begin{pmatrix} 1 & 0 & 0 & 0 \\ 0 & \frac{(e^{i\theta}+1)}{2} & \frac{(1+i)(e^{i\theta}-1)}{2\sqrt{2}} & 0 \\ 0 & \frac{(1-i)(e^{i\theta}-1)}{2\sqrt{2}} & \frac{(e^{i\theta}+1)}{2} & 0 \\ 0 & 0 & 0 & 1 \end{pmatrix}, \tag{12}
$$

where $e^{i\theta} = \cos(\theta) + i\sin(\theta)$. We put the detailed derivation of this gate in Apx. D. We then decide the connectivity of the circuit to calculate the size of the set of generators $\mathcal{G}$. For an $n$-qubit circuit, we have $P = n$ for Nearest Neighbor connectivity and $P = \frac{1}{2}n(n-1)$ for full connectivity. With the set of generators, we can compute the dimension of the corresponding DLA $\mathfrak{g}$. Note that The dimension of DLA is $d_k^2$ for both full connectivity (BS-full) and NN connectivity (BS-NN) for BS gate (how to connect the gate is shown in Fig. 1(b)), which indicates that BS gate is universal in the $d_k$-dimensional subspace (a possible decomposition for BS gate is in Fig. 7).

We can further utilize the overparameterization theory to determine the number of parameters or layers we need using BS gate. From Eq. 10, we can see that $M_c$ for BS gate with both connectivity is $d_k^2$, which can provide us with an approximation amount of parameters we need to build an ansatz. We are also able to estimate the number of layers we need, e.g. we need around $\lceil d_k^2/n \rceil$ layers for NN connectivity and $\lceil 2d_k^2/(n(n-1)) \rceil$ layers for full connectivity. It is vital for us to tell whether the ansatz have enough parameters and have reached its maximum capability.

## 3.4 UNITARY APPROXIMATION

To better illustrate the connection between the dimension of DLA and the above mentioned theories, we introduce the task of unitary approximation. As described in Sec. 3.1, we have linked the complete control of a quantum system to the unitary approximation. The unitary approximation aims to solve the problem of whether $\{\mathbf{U}(\boldsymbol{\theta})\}_{\boldsymbol{\theta}}$ is equal to $\mathcal{SU}(N)$ (see lemma. 3.1). For a target unitary matrix $\hat{\mathbf{U}}$ in $d_k$-dimensional HW subspace, the loss function for unitary approximation is

$$
\mathcal{L}_{UA}(\boldsymbol{\theta}) = 1 - |\operatorname{Tr}(\hat{\mathbf{U}}^\dagger \mathbf{U}(\boldsymbol{\theta}))|^2/d_k^2.
\tag{13}
$$

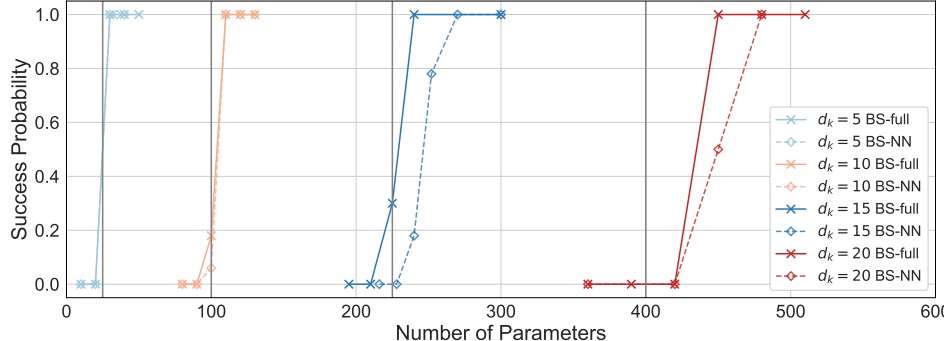

Figure 2: **Results for unitary approximation.** For each $d_k$, we randomly generate 50 unitary matrices based on Haar measure[1]. For each matrix, if the loss function $\mathcal{L} < 10^{-10}$, we say that we successfully approximate this matrix. The success probability indicates the portion of matrices we can approximate to $10^{-10}$ from all the 50 randomly generated matrices. The vertical line indicate the dimension of the DLA which is $d_k^2$. The results show that Nearest Neighbor (NN) connectivity and full connectivity for BS gate have similar performance with only slight difference.

We conduct the experiments on the challenging and widely accepted task of unitary approximation in the HW preserving subspace with dimension $d_k$. The purpose is to compare the performance of different gates and different connectivity. We first consider RBS gate. The DLA dimensions of RBS are less than $d_k^2$ for full connectivity (RBS-full) and Nearest Neighbor connectivity (RBS-NN). Theoretically the RBS gate can not form an ansatz that can approximate an arbitrary unitary matrix.

On the contrary, we can use BS gate to get complete control over the $d_k$-dimensional subspace. To better illustrate the theoretical results, we test RBS and BS gate on randomly generated unitary matrices. We set $d_k = \{\binom{5}{1}, \binom{5}{2}, \binom{6}{2}, \binom{6}{3}\} = \{5, 10, 15, 20\}$ and the results are shown in Fig. 2.

The results verify the theoretical analysis for both complete controllability and overparameterization phenomenon. RBS gate has failed all the tests thus we omit the RBS gates from Fig. 2. However, both connectivity for BS gate are able to approximate all the given matrices. Moreover, the experiments have shown a clear evidence that we need around $d_k^2$ parameters to approximate the unitary matrices. This verifies Eq. 10 that the dimension of the DLA for both connectivity of BS gate is $d_k^2$.

Note that the BS gate surpasses RBS gate on the unitary approximation task because of the dimension of DLA (according to Theorem 3.2, and only the dimension of BS gate reached $d_k^2$ ), but it is not necessarily a better choice on any problem. Unitary approximation is the hardest problem in $d_k$-dimensional subspace and if the problem is easy enough, RBS could also be a better choice. We will further illustrate this phenomenon in the numerical experiments.

### 3.5 THE TRAINABILITY OF THE HW PRESERVING ANSATZ

Now that we have the optimal control theory to determine whether the target state is within the reachable subspace of the ansatz, and the overparameterization theory to determine how many layers we need to construct a proper ansatz, we need to further discuss the trainability of the HW preserving ansatz to make sure that we can find a path to evolve the initial state to the target state without the presence of barren plateaus problem. A variational quantum circuit is said to have a barren plateaus problem if the gradients decay exponentially with the number of qubits (McClean et al., 2018). Thus, we have the following theorem:

**Theorem 3.5** *Consider a $n$-qubit quantum circuit operating in the subspace with Hamming Weight $k$. The variance of the cost function partial derivative is $Var_\theta[\partial_l C] \approx \frac{16k^2(n-k)^2}{n^4 d_k}$.*

Detailed proof of this theorem is provided in Apx. H. From the above results, we can further conclude that if $k$ is equal to 1, then $Var_\theta[\partial_l C] \approx \frac{16}{n^3}$. If $k = \frac{n}{2}$ on the other hand, $Var_\theta[\partial_l C] \approx$

---

[1]Utilizing qiskit.quantum_info.random_unitary in IBM Qiskit

$\binom{n}{n/2}^{-1}$, which is approximate to exponentially small. This result is consistent with the conjecture that the trainability of the circuit is closely related to $d_k$ and smaller $d_k$ will lead to better trainability.

## 4 EXPERIMENTS

We present results on two well-studied constrained VQAs: symmetry-preserving ground state energy estimation and feature selection. Experiments are performed on a machine with 190GB memory, one physical CPU with 32 cores AMD Ryzen 3970X CPU, 5 GPUs (NV GeForce RTX 3090).

### 4.1 SYMMETRY-PRESERVING STATE PREPARATION

**Background.** State preparation is a well-studied problem in quantum chemistry with ground state energy estimation as a popular application. The ground state of a molecule is its stationary state with the lowest allowed energy, i.e. ground state energy $E_0$, which can be estimated given the types and relative coordinates of its atoms as well as the number of orbitals and electrons. The energy is closely related to the molecular Hamiltonian $\mathcal{H}_m$, which is an Hermitian matrix embodying the energy of electrons and nuclei in the molecule. Given the final state $|\psi\rangle$ of the circuit, we have:

$$E_0 \leq \frac{\langle\psi|\,\mathcal{H}_m\,|\psi\rangle}{\langle\psi|\psi\rangle}, \tag{14}$$

where the equality holds if and only if $|\psi\rangle$ is the ground state. Notice that there are several well-defined symmetries from the physical level (Gard et al., 2020) such as the particle number, total spin and time-reversal, etc. Consider a molecule with $n$ orbitals and $k$ electrons, the ground state is bounded in the $d_k$-dimensional subspace.

**Experimental Setting.** We select three well-studied molecules, i.e. Hydrogen ($H_2$), Lithium Hydride (LiH), Water ($H_2O$) and Ammonia ($NH_3$). We obtain the Hamiltonian of these molecules from the Python package Open-Fermion (McClean et al., 2020). The computational basis for all the molecules is STO-3G. Detailed statistics of the molecules are listed in Tab. 1. LiH-8 is a simplified molecule from the

Table 1: Statistics of molecules. $n$ and $k$ are the number of orbitals and electrons respectively.

| Molecules | $H_2$ | LiH-8 | LiH | $H_2O$ | $NH_3$ |
|---|---|---|---|---|---|
| $n$ | 4 | 8 | 12 | 14 | 16 |
| $k$ | 2 | 2 | 4 | 10 | 10 |
| $d_k$ | 6 | 28 | 495 | 1001 | 8008 |

original LiH with only 8 orbitals and 2 electrons. To better illustrate the efficiency of the HW preserving ansatz, we select HEA ansatz (Kandala et al., 2017) and UCCSD (Romero et al., 2018) as the baselines. Among all the methods, only HEA acts in the whole Hilbert space and use $|\psi_0\rangle = 0^{\otimes n}$ as the initial state, all the HW preserving ansatz and UCCSD use Hartree-Fock state $|\psi_{HF}\rangle$ as the initial state. Thus, the loss for all the ansatz except HEA is:

$$\mathcal{L}_m(\boldsymbol{\theta}) = \langle\psi_{HF}|\,\mathbf{U}^\dagger(\boldsymbol{\theta})\mathcal{H}_m\mathbf{U}(\boldsymbol{\theta})\,|\psi_{HF}\rangle \tag{15}$$

**Results and Discussion.** We conduct two groups of experiments on 4 of the above mentioned molecules ($NH_3$ apart). We first use a fixed bond length and vary the number of layers. For all the molecules as shown in Fig. 3, our method can achieve an error below $1 \times 10^{-10}$ which is much smaller than the chemical accuracy at $1.6 \times 10^{-3}$Ha (we suspect the error at this level might be caused by the accuracy error of python). HEA is able to achieve comparable results with HW preserving ansatz on $H_2$, but with the problem size increases, the $2^n$ space for HEA to explore grows faster than the $d_k$-dimensional subspace. Thus, HEA will need much more parameters than HW preserving ansatz, which leads to the poor performance on LiH and $H_2O$.

Recall that we have three different dimensions of DLA for the HW preserving ansatz. The dimension of RBS with NN connectivity is $\frac{1}{2}n(n-1)$, RBS with full connectivity is $\frac{1}{2}d_k(d_k-1)$, and BS with both connectivity is $d_k^2$. From the results, we can see that RBS with NN connectivity is unable to solve the ground state energy estimation problem, and the other three can achieve consistent good estimation. We can thus conclude that the state preparation problem is much easier than the unitary approximation problem with the dimension of DLA be somewhere between $\frac{1}{2}n(n-1)$ and $\frac{1}{2}d_k(d_k-1)$. We can also observe from Fig. 3 that RBS gate with full connectivity performs better than BS gate with less parameters required to reach the exact energy. This reflects the importance of selecting a proper gate considering the physical qubit connectivity and problem difficulty.

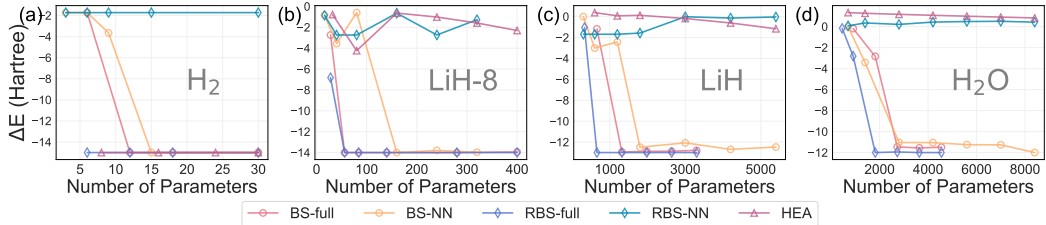

Figure 3: **The energy errors w.r.t. the number of parameters.** Note that HEA operates in the whole space, so it is possible for HEA to converge with more parameters. The number of parameters of UCCSD does not align with the circuit depth thus we omit the comparison in the figure.

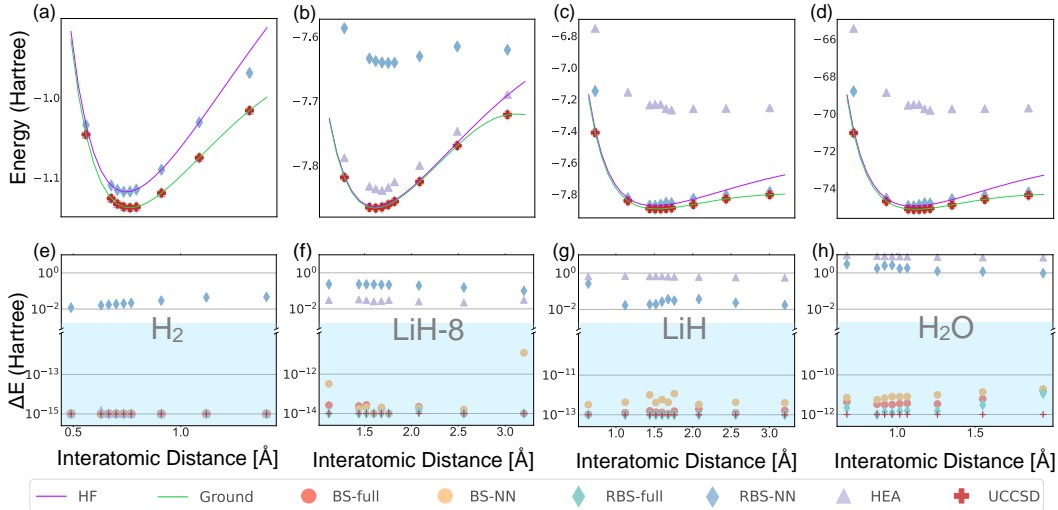

Figure 4: **Energy curve for different molecules.** (a-d) Potential energy curves of the bond distance. HF stands for the energy with Hartree-Fock state. Ground stands for the exact energy. (e-h) Absolute errors compared to the exact energy. Blue region indicates energy is within chemical accuracy.

As for $NH_3$, it is too large and takes unaffordable time to finish all the tests. We have the following data that using BS gate with full connectivity, we reach an energy error at $3 \times 10^{-10}$ with 21000 parameters. This verify the capability of BS gate to more complex molecules but the number of parameters required are indeed very large.

We then vary the bond distance of these molecules to see if the results are consistent on these methods. We fix the number of parameters for each method to the minimum value that reaches overparameterization based on the results in Fig. 3. From the results in Fig. 4, we can see that all the methods are consistent with different bond distances. We are able to provide comparable (or even better) results to UCCSD, which indicates that HW preserving ansatz is an efficient and steady substitution for HEA ansatz. The HW preserving gates can be decomposed into basic gates and the NN connectivity is the same as the connectivity of HEA. We can even adjust the connectivity based on the actual structure of the quantum hardware and provide ansatz with better capability, which is lower bounded by the NN connectivity ansatz.

## 4.2 Feature Selection

**Background.** We then study feature selection (Kira & Rendell, 1992; Kumar & Minz, 2014; Li et al., 2017), which has been widely used in machine learning, serving to mitigate the deleterious effects caused by cast dimension, simplify models for enhanced predictability, decrease redundant and irrelevant and speed up training. The feature selection problem can be mapped to a QUBO problem. Since Variational Quantum Eigensolvers can have potential advantages against classical solvers on QUBO problems (Tilly et al., 2022), numerous papers have utilized quantum algorithms to solve feature selection (Nguyen et al., 2014; Milne et al., 2017; Ferrari Dacrema et al., 2022).

Table 2: Statistics of the datasets for the task of feature selection.

| Dataset | Instances | Features | Feature Type |
|---|---|---|---|
| Wine Quality (Cortez et al., 2009) | 4898 | 11 | Real |
| Heart Disease (Janosi et al., 1988) | 303 | 13 | Categorical, Integer, Real |
| Titanic (Vanschoren et al., 2013) | 1045 | 14 | Boolean, Integer |
| Dry Bean (Koklu & Ozkan, 2020) | 13611 | 16 | Integer, Real |

Table 3: Results for the feature selection problem with best in bold. The numbers with striking lines denote over-selected feature dimensions due to the soft constraints on problem-solving.

| | BS | RBS | pyQUBO | | | | | | |
|---|---|---|---|---|---|---|---|---|---|
| | | | $\alpha = 0.5$ | $\alpha = 1$ | $\alpha = 5$ | $\alpha = 10$ | $\alpha = 50$ | $\alpha = 100$ | $\alpha = 1000$ |
| Wine Quality | **1.136342** | 1.135580 | ~~1.295212~~ | 1.080423 | 1.080423 | 1.099222 | 1.001571 | 0.983495 | 0.997417 |
| Heart Disease | **3.418328** | 3.418183 | ~~8.726707~~ | ~~5.635489~~ | 3.223445 | 2.963659 | 2.792978 | 2.697520 | 2.717723 |
| Titanic | **1.197799** | 1.197159 | ~~1.330309~~ | 0.962090 | 1.009051 | 0.936198 | 0.929956 | 0.916599 | 0.923464 |
| Dry Bean | 8.616844 | **8.616916** | ~~67.350723~~ | ~~20.765887~~ | 8.823824 | 7.790612 | 7.928807 | 7.864919 | 7.870827 |

**Experimental Setting.** One of the main methods to evaluate a feature selection algorithm is to utilize the mutual information (Guyon & Elisseeff, 2003) (see Apx. G for more details). We then use the following feature selection score to evaluate different algorithms:

$$S_{FS} = \mathbf{x}^\top \mathbf{Q} \mathbf{x}. \tag{16}$$

Thus, the target is to maximize the score function $S_{FS}$. However, selecting more than $k$ features can lead to larger scores than possible. To address the constraint, we add a penalty term to the information matrix $\mathbf{Q}$ in the QUBO problem as a soft constraint Hadfield et al. (2017):

$$\mathbf{x}^\top \mathbf{Q}' \mathbf{x} = \mathbf{x}^\top \mathbf{Q} \mathbf{x} - \alpha \Big( HW(\mathbf{x}) - k \Big)^2 \tag{17}$$

where $\alpha$ is the penalty coefficient and $k$ is the number of features to select. This constrained information matrix $\mathbf{Q}'$ can be converted into a Hamiltonian $\mathcal{H}_{FS}$ for the feature selection problem. The initial state should be an arbitrary state in the $d_k$-dimensional subspace. Notice that HW preserving ansatze do not need the penalty term so we can set $\alpha = 0$ for all the HW preserving ansatze.

We select four open-source datasets to test the HW preserving ansatz as well as a classical QUBO solver pyQUBO (Tanahashi et al., 2019; Zaman et al., 2021). Statistics for the datasets are listed in Tab. 2. We set $k$ for all the datasets as 3 indicating that we select the top 3 features from all the datasets, which can also lead to various $d_k$.

**Results.** The results are listed in Tab. 3, where some numbers are stroked for denoting "invalidity" as they are obtained with more than $k$ features. This also indicates that the penalty term is not large enough to ensure legal results. The results show that the selection of $\alpha$ is important in the large $2^n$ Hilbert space. A small $\alpha$ will lead to illegal states and an overlarge $\alpha$ can lead to hard optimization landscape. Both HW preserving ansatz can achieve better results than the soft constrained QUBO solver, which again verifies the supremacy of searching in a hard constrained subspace.

## 5 CONCLUSION AND LIMITATION

This paper mainly focus on the symmetries and constraints in the VQAs, which lead to a smaller subspace, namely a $d_k$-dimensional subspace, as a coherent fit for the HW preserving ansatz. Thus, we revisit the HW preserving ansatz and link the ansatz with these constrained VQAs. We also provide theoretical analysis on the capability and expressivity of the HW preserving ansatz and verify these theoretical results on unitary approximation problem. With these theories, we can generate a new gate that is universal in the subspace. We conduct numerical experiments on ground state energy estimation and feature selection problems, both with superior performance.

The main limitation of this paper is that the BS gate proposed in this paper is quite deep if we decompose the gate with basic gates. It is possible that there is a perfect HW preserving gate which is universal in the subspace and also easy to decompose. We have provided all the theories needed to verify the gate, but finding and decomposing the gate is still remains an open question.

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

# A RELATED WORKS

**Symmetries and Constraints in Quantum Systems.** We first focus on the quantum chemistry problem where we have several well-defined symmetries to illustrate the necessity of considering hard constraints. The VQE algorithm on ground state energy estimation problems can be divided into two main categories. One is based on chemical properties such as the unitary coupled cluster method (Bartlett et al., 1989; Ryabinkin et al., 2018; Shkolnikov et al., 2023) and utilizes Trotter decomposition to form the ansatz. This approach will lead to deep circuits which is unrealizable on current hardware. The other approach does not follow these chemical properties and only uses parameterized gates that are available on the processor such as the Hardware Efficient Ansatz (HEA) (Kandala et al., 2017; Bian et al., 2019) and the Quantum Architecture Search (QAS) (Wu et al., 2023) methods. These methods can produce relatively shallow and feasible ansatz, but they can also generate unphysical states. The most important symmetries (or invariants) in the quantum chemistry problem are the particle number and total spin (Gard et al., 2020). This kind of symmetry is widely used in various problems such as VQE (Gard et al., 2020), Feature selection (Nembrini et al., 2021; Mücke et al., 2023), constrained QUBO (Hadfield et al., 2019), etc.

**Previous HW preserving related works.** For the above-mentioned constraints, we can easily address them by adding regularizers in the Hamiltonian to give a penalty if we break the constraints. However, this method can only provide soft constraints and we can still have illegal results in the final states. Thus, finding an ansatz that only operates in the constrained subspace can guarantee the results are admissible. There are several works that are related to the idea of HW preserving to enable hard constraints on the circuits.

The HW preserving gate is previously used in several literature where (Kerenidis et al., 2021) first proposed the idea of quantum orthogonal neural networks. The authors proposed RBS gate with a pyramid structure on the quantum circuit and called it the Pyramidal Circuit. This circuit can act as a substitution for the classical orthogonal neural networks. The Pyramidal Circuit is used as a data loader with experiments on image classification for MNIST and medical images (Landman et al., 2022). The quantum orthogonal layer can be taken as a special case for the HW preserving circuit with $k = 1$.

Cherrat et al. (2023) further utilizes the RBS gate to solve the hedging problem. This paper investigates the quantum orthogonal layer as well as the quantum compound layer. The quantum orthogonal layer is used as an encoding layer with different architectures of RBS gates arrangement. Results on financial problems especially the hedging problem on a trapped-ion quantum processor are provided in the paper. The authors also provide theoretical analysis that the variance of the gradients decays only polynomially with the number of qubits, which indicates the HW preserving circuit can have better trainability than HEA.

Notice that there is a concurrent work Monbroussou et al. (2023), which shared a similar thought with us by utilizing DLA to analyze the trainability and expressivity of the HW preserving ansatz. They mainly focused on building quantum data loader with HW preserving ansatz and the two proposed HW preserving gates RBS and FBS are able to fulfill the task. To compare Monbroussou et al. (2023) and this work, we have the following differences. (1) We analyze the expressivity from a more general perspective instead of only as a dataloader; (2) we introduce the overparameterization theory to figure out the number of parameters needed in the ansatz; (3) we focus more on constrained VQE problems as Monbroussou et al. (2023) focus on building dataloader for supervised quantum deep learning models; (4) we provide a more thorough background check for HW preserving ansatz by introducing other gates such as XY-mixer which is widely used in constrained QAOA.

These above papers share an opinion to use HW preserving gate as a data loader for quantum deep learning. It is obvious that HW preserving ansatz can only load data in the $d_k$-dimensional subspace. For a deep learning problem, it's rare to find all the data in the dataset falling within such a limited subspace. Furthermore, when additional quantum layers are added to the circuit, the circuit no longer maintains its HW preserving property, which negates the advantages of using the HW preserving ansatz.

Though not using the name of HW preserving ansatz, there are also works utilizing specific gates to replace the mixing operator in QAOA so that they can tackle a certain kind of constraint (Hadfield et al., 2019). The proposed XY-mixer as well as some other mixers in (Hadfield et al., 2019) can

be classified as special kinds of HW preserving ansatz. However, those gates in (Hadfield et al., 2019) are found to have HW preserving properties, and we would like to know if we can generate HW preserving gates with thorough understanding on the properties of those generated gates. Thus, we rethink the HW preserving ansatz and link this ansatz to constrained VQEs to enlarge the usage of HW preserving ansatz.

**Existing gates that are HW preserving.** Apart from the aforementioned RBS gate that we have discussed in detail in the paper, we also have several HW preserving gates from previous literature. Hadfield et al. (2019) proposed a quantum alternating operator ansatz which utilize the XY-mixer and RZZ gate to form a constrained quantum ansatz. The XY-mixer is a HW preserving gate with its Hamiltonian constructed by XX and YY rotation $H_{XY} = X \otimes X + Y \otimes Y$. The Hamiltonians for XX, YY, and ZZ gates are

$$\mathbf{H}_{XX} = \begin{pmatrix} 0 & 0 & 0 & 1 \\ 0 & 0 & 1 & 0 \\ 0 & 1 & 0 & 0 \\ 1 & 0 & 0 & 0 \end{pmatrix}, \mathbf{H}_{YY} = \begin{pmatrix} 0 & 0 & 0 & -1 \\ 0 & 0 & 1 & 0 \\ 0 & 1 & 0 & 0 \\ -1 & 0 & 0 & 0 \end{pmatrix}, \mathbf{H}_{ZZ} = \begin{pmatrix} 1 & 0 & 0 & 0 \\ 0 & -1 & 0 & 0 \\ 0 & 0 & -1 & 0 \\ 0 & 0 & 0 & 1 \end{pmatrix}.$$
(18)

Thus, we can see that XX, YY, and ZZ gates are all HW preserving gates, and we might use XX, YY, and ZZ to construct other HW preserving gates. However, if we consider the general form of HW preserving gate in Eq. 11, we can only construct HW preserving gates with $a = c$ and $b = \bar{b}$. That is exactly why we need a more general form to construct and analyze the HW preserving gates so that we can fully understand the HW preserving circuit with the ability to generate any specific gates we want.

In line with the RBS gate, we here provide a brief analysis on the XY-mixer utilizing all the theoretical tools provided in the paper. First of all, the hamiltonian and unitary matrix of the XY-mixer is

$$\mathbf{H}_{XY} = \begin{pmatrix} 0 & 0 & 0 & 0 \\ 0 & 0 & 1 & 0 \\ 0 & 1 & 0 & 0 \\ 0 & 0 & 0 & 0 \end{pmatrix}, \quad \mathbf{U}_{XY} = \begin{pmatrix} 1 & 0 & 0 & 0 \\ 0 & \cos(\theta) & -i\sin(\theta) & 0 \\ 0 & -i\sin(\theta) & \cos(\theta) & 0 \\ 0 & 0 & 0 & 1 \end{pmatrix}.$$
(19)

For the NN connectivity, the dimension of DLA of the XY-mixer is

$$dim(\mathfrak{g}_{XY}) = \begin{cases} (n+1)(n-1) & n \text{ is odd} \\ \frac{1}{2}n(n-1) & n \text{ is even} \end{cases}$$
(20)

For the FC connectivity, the dimension of DLA of the XY-mixer is

$$dim(\mathfrak{g}_{XY}) = \begin{cases} (d_k+1)(d_k-1) & n \neq 2k \\ \frac{1}{2}(d_k+2)(d_k-2) & n = 2k \end{cases}$$
(21)

Thus, we can conclude that XY-mixer with full connectivity is quite capable of solving most of the problems except for some rare cases and maybe that is exactly why it is so famous for solving constrained QAOA. However, the XY-mixer on NN-connectivity is not very good even with the RZZ gate as the phase operator (the dimension of DLA is still not full). Therefore, we can conclude that the XY-mixer might not be so capable considering the physical qubit topology. One possible solution to this is to combine other gates to increase the initial generators and gain full controllability under both connectivities.

# B FURTHER INTRODUCTION ON PRELIMINARIES

## B.1 QUANTUM COMPUTING AND QUANTUM MACHINE LEARNING

In quantum computing, 'qubit' (abbreviation of 'quantum bit') is a key concept which is similar to a classical bit with a binary state. The two possible states for a qubit are the state $|0\rangle$ and $|1\rangle$, which correspond to the state 0 and 1 for a classical bit respectively. We refer the readers to the

textbook (Nielsen & Chuang, 2002) for comprehension of quantum information and quantum computing. Here we give a brief introduction to the background.

A quantum state is commonly denoted in bracket notation. It is also common to form a linear combination of states, which we call a superposition: $|\psi\rangle = \alpha|0\rangle + \beta|1\rangle$. Formally, a quantum system on $n$ qubits is an $n$-fold tensor product Hilbert space $\mathcal{H} = (\mathbb{C}^2)^{\otimes d}$ with dimension $2^d$. For any $|\psi\rangle \in \mathcal{H}$, the conjugate transpose $\langle\psi| = |\psi\rangle^\dagger$. The inner product $\langle\psi|\psi\rangle = ||\psi||_2^2$ denotes the square of the 2-norm of $\psi$. The outer product $|\psi\rangle\langle\psi|$ is a rank 2 tensor. Computational basis states are given by $|0\rangle = (1,0)$, and $|1\rangle = (0,1)$. The composite basis states are defined by e.g. $|01\rangle = |0\rangle \otimes |1\rangle = (0,1,0,0)$.

Analog to a classical computer, a quantum computer is built from a quantum circuit containing wires and elementary quantum gates to carry around and manipulate the quantum information. These gates can be parameterized quantum gates such as $Rx(\theta), Ry(\theta), Rz(\theta)$ or basic gates as $\sigma_x, \sigma_y, \sigma_z, CNOT, CZ$. For an initial state $|\psi_0\rangle$ and $L$ layers of quantum circuit, the final state $|\psi'\rangle$ can be denoted as

$$|\psi'\rangle = \prod_{l=1}^{L} \mathbf{U}_l |\psi_0\rangle. \tag{22}$$

### B.2 Hamming Distance and Hamming Weight

In this section, we recall the definition of Hamming distance and Hamming weight, which is constantly used in this paper. Considering two binary vectors $\mathbf{a}$ and $\mathbf{b}$ with $\mathbf{a}, \mathbf{b} \in \{0,1\}^N$, where $N$ is the dimension of the vectors. We can define the corresponding Hamming distance of these two vectors as follows:

**Definition B.1** *Hamming distance: The Hamming distance $\mathcal{D}$ of two binary vectors $\mathbf{a}$ and $\mathbf{b}$ is:*

$$\mathbf{c} = \mathbf{a} \oplus \mathbf{b},$$
$$\mathcal{D}(\mathbf{a}, \mathbf{b}) = \sum_{i=1}^{N} \mathbf{c}_i, \tag{23}$$

*where $\oplus$ stands for exclusive OR operator.*

With the definition of Hamming distance, we can further define the Hamming weight of a given binary vector $\mathbf{a}$.

**Definition B.2** *Hamming weight: Let $\mathbf{0} = 0^N$. The Hamming weight $HW$ of binary vector $\mathbf{a}$ is:*

$$HW(\mathbf{a}) = \mathcal{D}(\mathbf{a}, \mathbf{0}), \tag{24}$$

## C Detailed algorithm for computing the dimension of DLA

We define a transformation $\sigma$ which satisfies $\mathbf{h} = \sigma(\mathbf{H})$ ($\mathbf{H} \in \mathbb{C}^{N \times N}, \mathbf{h} \in \mathbb{C}^{N^2 \times 1}$), where $\mathbf{h}$ is a column vector obtained by concatenating the columns of $H$ vertically. $\sigma^{-1}$ stands for the inversed transformation which maps a column vector $\in \mathbb{C}^{N^2 \times 1}$ back to a matrix $\in \mathbb{C}^{N \times N}$. $\mathbf{D}_{:,j}$ denotes the $j$-th column of matrix $\mathbf{D}$, and rank$(\mathbf{D}, \mathbf{h})$ is the rank of matrix $\mathbf{D}$ appending column vector $\mathbf{h}$ to the right. $[\cdot, \cdot]$ denotes the commutator between two matrices with $[\mathbf{A}, \mathbf{B}] = \mathbf{AB} - \mathbf{BA}$. The pseudo code for the algorithm is shown in Alg. 1.

## D Converting Hamiltonian to Unitary Matrix

$$\mathbf{H}_{BS} = \begin{pmatrix} 0 & 0 & 0 & 0 \\ 0 & \frac{1}{2} & \frac{1+i}{2\sqrt{2}} & 0 \\ 0 & \frac{1-i}{2\sqrt{2}} & \frac{1}{2} & 0 \\ 0 & 0 & 0 & 0 \end{pmatrix} \tag{25}$$

---

**Algorithm 1** Computing the dimension of DLA

---

**Require:** Generator set $\mathcal{G} = \{\mathbf{H}_1, \mathbf{H}_2, \cdots, \mathbf{H}_P\}$, $\mathbf{h}_i = \sigma(\mathbf{H}_i)$.
  let $\mathbf{D} = \mathbf{h}_1$;
  let $r = \text{rank}(\mathbf{D})$;
  **for** $i = 2$ **to** $P$ **do**
    **if** $\text{rank}(\mathbf{D}, \mathbf{h}_i) > r$ **then**
      $\mathbf{D}$.append($\mathbf{h}_i$);
      $r = r + 1$;
    **end if**
  **end for**
  let $r_{out} = 0$;
  **while** $r_{out} \neq r$ **and** $r \neq N^2$ **do**
    **for** $l = r_{out} + 1$ **to** $r$ **do**
      **for** $j = 1$ **to** $l$ **do**
        let $\mathbf{H}_{tmp} = [\sigma^{-1}(\mathbf{D}_{:,l}), \sigma^{-1}(\mathbf{D}_{:,j})]$, $\mathbf{h}_{tmp} = \sigma(\mathbf{H}_{tmp})$;
        **if** $\text{rank}(\mathbf{D}, \mathbf{h}_{tmp}) > r$ **then**
          $\mathbf{D}$.append($\mathbf{h}_{tmp}$);
          $r = r + 1$
        **end if**
      **end for**
    **end for**
    let $r_{out} = r$;
    let $r = \text{rank}(\mathbf{D})$;
  **end while**
  **return** $r_{out}$

---

We can get the unitary matrix of the BS gate by the Taylor expansion:

$$\mathbf{U}_{BS}(\theta) = e^{i\theta\mathbf{H}_{BS}} = \sum_{j=0}^{\infty} \frac{1}{j!}(i)^j(\theta)^j\mathbf{H}_{BS}^j. \tag{26}$$

When we design the Hermitian matrix $\mathbf{H}_{BS}$ for the BS gate, we set

$$\mathbf{H}_{BS}^2 = \mathbf{H}_{BS}. \tag{27}$$

Thus, we have a special character for the matrix $\mathbf{H}_{BS}$ that

$$\mathbf{H}_{BS}^j = \mathbf{H}_{BS}^{j-1} = ... = \mathbf{H}_{BS}^2 = \mathbf{H}_{BS}. \tag{28}$$

The unitary matrix can be simplified as follows:

$$\mathbf{U}_{BS}(\theta) = \mathbf{I} + \mathbf{H}_{BS}\sum_{j=1}^{\infty}(i\theta)^j = \mathbf{I} + \sum_{j=1}^{\infty}i(-1)^{j-1}\frac{(\theta)^{2j-1}}{(2j-1)!} + \sum_{j=1}^{\infty}(-1)^j\frac{(\theta)^{2j}}{(2j)!}, \tag{29}$$

We Use the Taylor expansion of $\sin(\theta), \cos(\theta)$ at $\theta = 0$

$$\mathbf{U}_{BS}(\theta) = \mathbf{I} + \mathbf{H}_{BS}(i\sin\theta + \cos\theta - 1) \tag{30}$$

Therefore, the unitary matrix for BS is

$$\mathbf{U}_{BS}(\theta) = \begin{pmatrix} 1 & 0 & 0 & 0 \\ 0 & \frac{(\cos\theta + i\sin\theta + 1)}{2} & \frac{(1+i)(\cos\theta + i\sin\theta - 1)}{2\sqrt{2}} & 0 \\ 0 & \frac{(1-i)(\cos\theta + i\sin\theta - 1)}{2\sqrt{2}} & \frac{(\cos\theta + i\sin\theta + 1)}{2} & 0 \\ 0 & 0 & 0 & 1 \end{pmatrix}. \tag{31}$$

## E   REMARKS ON THE BIT-FLIP ERROR

It has occurred to us that the bit flip error can easily destroy the Hamming Weight of the quantum states and thus affect the subspace that we are operating in. However, this problem can be tackled

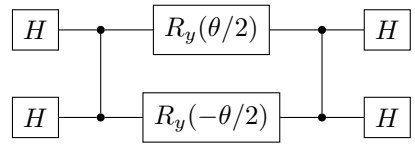

Figure 5: A decomposition of RBS($\theta$) gate.

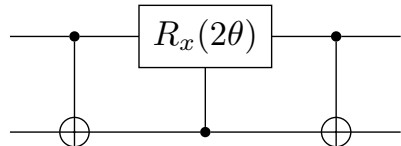

Figure 6: A decomposition of XY-mixer($\theta$) gate. Notice that controlled-Rx gate requires at least one two-qubit gate to implement.

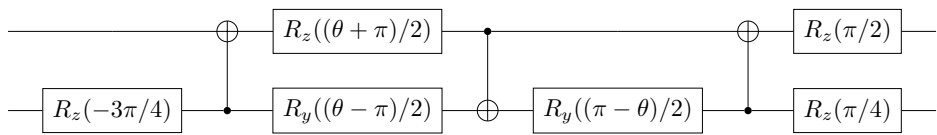

Figure 7: A possible decomposition of BS($\theta$) gate.

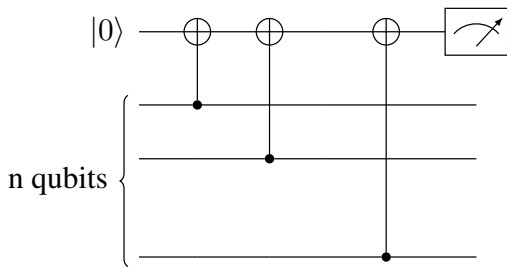

Figure 8: Circuit for the parity check.

by utilizing a parity check to make the whole HW preserving circuit bit-flip fault tolerant. The Hamming weight $k$ indicates that we know whether there are odd number or even number of 1s in the state. We can utilize one ancilla qubit as shown in Fig. 8 to enable the parity check. The ancilla qubit is set to $|0\rangle$ in the beginning and all the working qubits are linked to the ancilla qubit with CNOT gate, with the ancilla qubit as the target bit. If $k$ is odd, then the measurement outcome of the ancilla qubit should be $|1\rangle$. Otherwise, there is a bit flip occurring in the working qubits. Thus, it is much easier to detect a bit-flip error on the HW preserving circuit compared to Shor's code with two ancilla qubits required to protect one qubit from bit-flip error. We can either rerun the circuit or locate the qubit to correct the bit-flip error.

## F    FURTHER RESULTS ON UNITARY APPROXIMATION

In this section, we provide further results on the unitary approximation problem. To better demonstrate the convergence of the algorithms with different number of layers, we list the following results in Fig. 9. We also include RBS gate to better illustrate the best precision RBS can get. It is clear that no matter how many parameters we add, RBS gate is unable to go further than $10^{-3}$. From the comparison between the first column and the second column, we can see that NN connectivity requires more iterations to get similar results compared to ull connectivity. The results on BS also

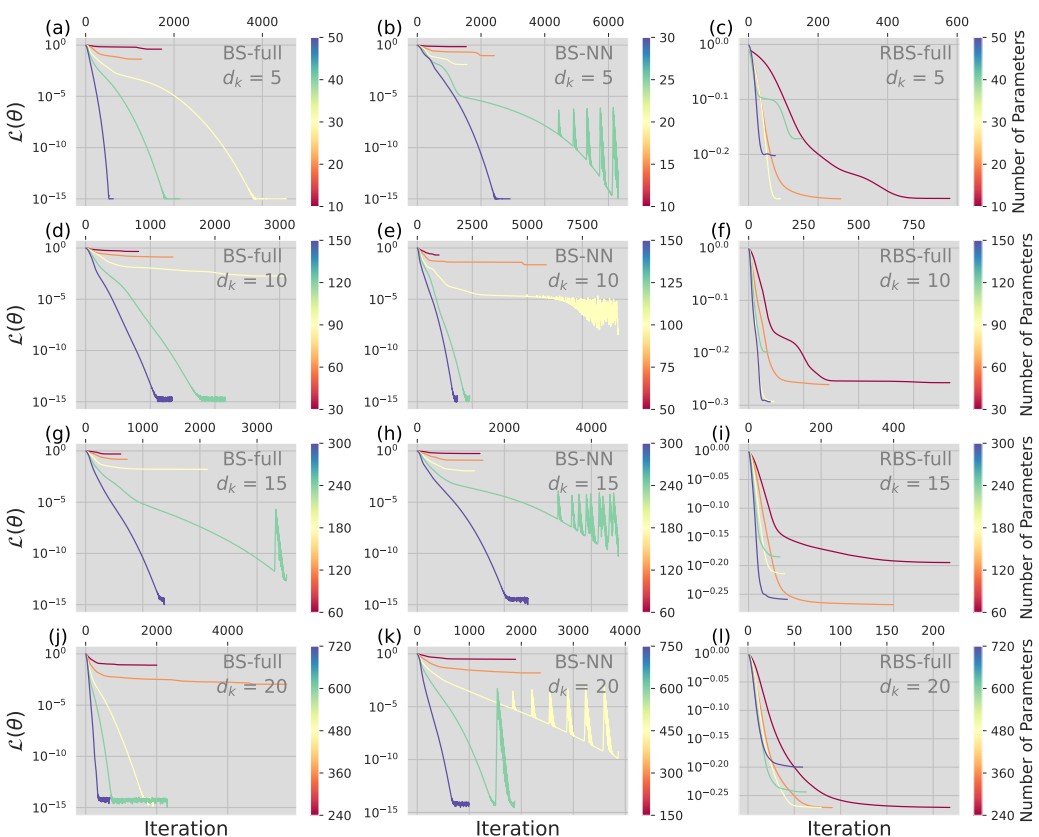

Figure 9: **The energy errors w.r.t. iterations with the different number of parameters.**

implies the overparameterization phenomenon. We need more than $d_k^2$ parameters to reach $10^{-15}$, and further adding parameters will not lead to better performance but less iterations.

# G   FILTER METHODS AND MUTUAL INFORMATION

Out of three primary categories of feature selection techniques, which are wrapper methods, filter methods and embedded methods, we choose filter methods to address the task. Filter methods are particularly advantageous for their ability to execute swiftly and independently on quantum circuits. Our chosen filter method is based on information theory, specifically utilizing mutual information to quantify the relativity between individual features and the target variable. A higher mutual information indicates that the selected feature has a greater predictive power for the target. The mutual information between feature $X$ and target $Y$ is in the form:

$$
\begin{aligned}
I(X;Y) &= D_{KL}(P_{X,Y} \parallel P_X \otimes P_Y) \\
&= \sum_{y \in Y} \sum_{x \in X} P_{X,Y}(x,y) \log(\frac{P_{X,Y}(x,y)}{P_X(x)P_Y(y)}) \\
&= H(X) + H(Y) - H(X,Y)
\end{aligned}
\tag{32}
$$

And we apply conditional mutual information to quantify the relationship between a specific feature, denoted as $X$, and the target variable on condition of another feature $Z$. A higher value of conditional mutual information signifies that the feature $X$ exhibits a greater degree of relevance to target and is independent of $Z$. Conversely, if the conditional mutual information is lower, it shows that the relativity between $Z$ and the target is primarily mediated by the presence of $Z$, and thus, $Z$ may be considered a redundant variable if we already picked $Z$. The conditional mutual information is in the form:

$$
\begin{aligned}
I(X;Y|Z) &= \mathbb{E}_Z[D_{KL}(P_{(X,Y)|Z} \parallel P_{X|Z} \otimes P_{Y|Z})] \\
&= \sum_{z \in Z} \sum_{y \in Y} \sum_{x \in X} P_{X,Y,Z}(x,y,z) \log(\frac{P_Z(z)P_{X,Y,Z}(x,y,z)}{P_{X,Z}(x,z)P_{Y,Z}(y,z)}) \\
&= H(X|Z) + H(Y|Z) - H(X,Y|Z)
\end{aligned}
\tag{33}
$$

With the acquisition of mutual information and conditional mutual information, we can encode all the n features into a binary vector $\boldsymbol{x} := x_1, ..., x_n \in \{0,1\}^n$ where $x_i$ signifies the $i$-th feature with 1 means the feature is chosen. Subsequently, this information can be amalgamated into a $n \times n$ matrix $Q$ where the items on the diagonal $Q_{ii} := -I(X_i; Y)$ correspond to the mutual information between the $i$-th feature $X_i$ and target $Y$ and the off-diagonal elements $Q_{ij} := -I(X_i; Y|X_j)$. Then we reformulate the feature selection problem, seeking to maximize both mutual information and conditional mutual information as well as considering the penalty, into the form of a new QUBO problem $\mathbf{Q}'$, and add the soft constrain Hadfield et al. (2017):

$$
\mathbf{x}^\top \mathbf{Q}' \mathbf{x} = \mathbf{x}^\top \mathbf{Q} \mathbf{x} - \alpha \Big( HW(\mathbf{x}) - k \Big)^2
\tag{34}
$$

$$
\boldsymbol{x}^* = \arg\max_{\boldsymbol{x} \in \{0,1\}^n} \left( \boldsymbol{x}^\top Q \boldsymbol{x} - \alpha \Big( HW(\boldsymbol{x}) - k \Big)^2 \right) = \arg\max_{\boldsymbol{x} \in \{0,1\}^n} \left( \boldsymbol{x}^\top Q' \boldsymbol{x} \right)
\tag{35}
$$

# H   DETAILED PROOF FOR THE TRAINABILITY

**Theorem H.1** *Consider a $n$-qubit quantum circuit operating in the subspace with Hamming Weight $k$. The variance of the cost function partial derivative is $Var_\theta[\partial_l C] \approx \frac{16k^2(n-k)^2}{n^4 d_k}$.*

$Proof.$ Consider the partial derivative of the cost function $C$ with respect to the parameters $\theta$. For some parameter $\theta_l$ in the $l$-th RBS gate, we have:

$$\partial_l C(\theta) = \partial_l(Tr[U(\theta)\rho U(\theta)^\dagger O])$$
$$= \partial_l(Tr[U_-\rho U_-^\dagger O_+])$$
$$= iTr[U_-\rho U_-^\dagger [H_l, O_+]]$$

where $\rho$ is the input state, $O$ is the observable to measure, $U_-$ denotes the unitary matrix of the circuit before the $l$-th gate and $U_+$ denotes the unitary matrix after gate $l$. $O_+ = U_+^\dagger O U_+$. The variance of the partial derivative is

$$Var_\theta[\partial_l C] = \int_{\mathcal{U}_+} dU_+ \int_{\mathcal{U}_-} dU_- (\partial_l C(\theta))^2$$
$$= \int_{\mathcal{U}_+} dU_+ \int_{\mathcal{U}_-} dU_- (iTr[U_-\rho U_-^\dagger [H_l, O_+]])^2,$$

where $[\cdot, \cdot]$ denotes the commutator of two matrices.

$$Var_\theta[\partial_l C] = -\int_{\mathcal{U}_+} dU_+ (\frac{Tr[\rho^2]Tr[[H_l, O_+]^2]}{d_k^2 - 1} - \frac{Tr^2[\rho]Tr[[H_l, O_+]^2]}{d_k(d_k^2 - 1)})$$
$$= -\int_{\mathcal{U}_+} dU_+ (Tr[[H_l, O_+]^2]\frac{d_k * Tr[\rho^2] - Tr^2[\rho]}{d_k(d_k^2 - 1)})$$
$$= -\frac{d_k * Tr[\rho^2] - Tr^2[\rho]}{d_k(d_k^2 - 1)} \int_{\mathcal{U}_+} dU_+ Tr[[H_l, O_+]^2].$$

The initial state $|\psi_0\rangle$ is set to be the uniform superposition over all computational basis in the $d_k$ subspace. Thus, we have $Tr[\rho] = 1, Tr[\rho^2] = 1$.

$$Var_\theta[\partial_l C] = -\frac{1}{d_k(d_k + 1)} \int_{\mathcal{U}_+} dU_+ Tr[[H_l, O_+]^2]$$
$$= -\frac{2}{d_k(d_k + 1)}[Tr[H_l O_+ H_l O_+] - Tr[H_l H_l O_+ O_+]]$$
$$= -\frac{2}{d_k(d_k + 1)}(\frac{Tr[H_l^2]Tr^2[O]}{d_k^2 - 1} - \frac{Tr[H_l^2]Tr[O^2]}{d_k(d_k^2 - 1)} - \frac{Tr[H_l^2]Tr[O^2]}{d_k})$$
$$= -\frac{2Tr[H_l^2]}{d_k(d_k + 1)}(\frac{Tr^2[O] - d_k Tr[O^2]}{d_k^2 - 1}),$$

where $Tr[H_l^2] = 2\binom{n-2}{k-1} = \frac{2k(n-k)}{n(n-1)}d_k$. We take $Z_0$ as the observable, then $Tr[O] = \frac{d_k(n-2k)}{n}, Tr[O^2] = d_k$. (other observables will also hold with the same magnitude) Thus, we have

$$Var_\theta[\partial_l C] = -\frac{2}{d_k(d_k + 1)} \times \frac{2k(n-k)d_k}{n(n-1)} \times (\frac{\frac{d_k^2(n-2k)^2}{n^2} - d_k^2}{d_k^2 - 1})$$
$$= \frac{4k(n-k)}{(d_k + 1)n(n-1)} \times \frac{d_k^2(n^2 - (n - 2k)^2)}{(d_k^2 - 1)n^2}$$
$$= \frac{4k(n-k)}{(d_k + 1)n(n-1)} \times \frac{d_k^2(4nk - 4k^2)}{(d_k^2 - 1)n^2}$$
$$= \frac{16k^2(n-k)^2 d_k^2}{(d_k + 1)n^3(n-1)(d_k^2 - 1)} \approx \frac{16k^2(n-k)^2}{n^4 d_k}.$$

We can further analyze that if the $k$ is only 1, then $Var_\theta[\partial_l C] \approx \frac{16}{n^3}$. If the $k = \frac{n}{2}$ on the other hand, $Var_\theta[\partial_l C] \approx \binom{n}{n/2}^{-1}$, which is approximate to exponentially small. This result is consistent with the conjecture that the trainability of the circuit is closely related to $d_k$ and smaller $d_k$ will lead to better trainability.

