# OpenReview forum: "Rethinking the symmetry-preserving circuits for constrained variational quantum algorithms"
_ICLR.cc/2024/Conference — ICLR 2024 poster_

### Official Review · Reviewer_obpr · 2023-10-19

**Soundness:** 4 excellent
**Presentation:** 3 good
**Contribution:** 2 fair
**Rating:** 5
**Confidence:** 4

**Summary:**

The authors review the known Hamming Weight (HW) preserving ansatz and discuss how it fits into the dynamical Lie algebra framework of (Larocca et al., 2023). The authors give an elementary gate set that they show generates all HW preserving gates (i.e., that gives "full controllability"). The authors then perform numerics demonstrating the practical utility of the HW preserving ansatz.

**Strengths:**

The authors give a nice review on the HW preserving ansatz and link it to the well-known line of results discussing when quantum machine learning (QML) models are trainable by examining their controllability. The authors also describe a set of gates to parameterize the HW preserving ansatz in a controllable way.

**Weaknesses:**

Most of the paper is review. For instance, arXiv:2303.16585 (which the authors cite) already demonstrates that the HW preserving ansatz is efficiently trainable. Though never explicitly linked to (Larocca et al., 2023) in arXiv:2303.16585, the intuition that the trainability comes from the large symmetry of the ansatz is stated and described: see, e.g., their Figure 3.

**Questions:**

What are some advantages to directly linking trainability to subspace controllability given the trainability of the HW preserving ansatz was already known?

---

> ### Author Response · Authors · 2023-11-16
>
> >***W1: Most of the paper is review. For instance, arXiv:2303.16585 (which the authors cite) already demonstrates that the HW preserving ansatz is efficiently trainable. Though never explicitly linked to (Larocca et al., 2023) in arXiv:2303.16585, the intuition that the trainability comes from the large symmetry of the ansatz is stated and described: see, e.g., their Figure 3.***
>
> **Ans:** Thanks for your comments.
>
> **1. We would like to first address the difference between our work and [1].**
>
> [1] proposed HW preserving circuit using existing gates (such as RBS and FBS) as orthogonal layers for quantum kernels and analyze the trainability of RBS gate. Our work, on the other hand, focuses on fully understanding the HW preserving gates so that we can construct an HW preserving gate as well as build an HW preserving ansatz. Furthermore, we focus on analyzing the capability and expressivity by giving the readers some guidance on how many layers we need to solve a certain problem. We omit the analysis of the trainability since we believe [1] has already done that. But thanks for the comments by Reviewer ADjN, we carefully reviewed the results and proof in [1] and we are surprised to find it incorrect. There are still chances that we misunderstood the calculation for the variance bound in [1], so we decided to give our own results here with a sketch proof. Detailed derivation will be included in the revision.
>
> We first discuss the results shown in [1]. We tried the derivation of theorem 2 and find that $\partial_lC_k=\langle\psi_0|(\sigma_X^i\otimes\sigma_X^j+\sigma_Y^i\otimes\sigma_Y^j)|\psi_0\rangle$ should be $1$ instead of $\frac{1}{4}$. For the initial state in the HW k subspace, the results in [1] are $\partial_lC_k=\langle\psi_k|(\sigma_X^i\otimes\sigma_X^j+\sigma_Y^i\otimes\sigma_Y^j)|\psi_k\rangle=2(\frac{2\tbinom{n-2}{k-1}}{\tbinom{n}{k}})^2=\Omega(\frac{1}{n^6})$, where according to our derivation, the results should be $\partial_lC_k=\langle\psi_k|(\sigma_X^i\otimes\sigma_X^j+\sigma_Y^i\otimes\sigma_Y^j)|\psi_k\rangle=2\frac{2\tbinom{n-2}{k-1}}{\tbinom{n}{k}}$, $\frac{\tbinom{n-2}{k-1}}{\tbinom{n}{k}}=\frac{k(n-k)}{n(n-1)}$. Even if the original result is correct that $\partial_lC_k=8(\frac{k(n-k)}{n(n-1)})^2$, $\partial_lC_k=\mathcal{O}(1)$ if we take $k=n/2$, which seems incorrect.
>
>
> Now we provide the results based on our derivation. Without loss of generality, we take RBS gate as an example since [1] also derives the bound based on RBS gate.
>
>
> ***Theorem 1. Consider a quantum circuit operating in the subspace with Hamming Weight $k$. The variance of the cost function partial derivative is $$Var_\theta[\partial_lC]\approx \frac{16k^2(n-k)^2}{n^4d_k}$$***
>
> *Proof.* Consider the partial derivative of the cost function $C$ with respect to the parameters $\theta$. For some parameter $\theta_l$ in the $l$-th RBS gate, we have:
> $$\partial_{l} C(\theta)=\partial_l(Tr[U(\theta)\rho U(\theta)^{\dagger}O])$$
> $$=\partial_l(Tr[U_-\rho U_-^{\dagger} O_+]) $$
> $$=iTr[U_-\rho U_-^{\dagger} [H_l,O_+]],$$
>
> where $\rho$ is the input state, $O$ is the observable to measure, $U_-$ denotes the unitary matrix of the circuit before the $l$-th gate and $U_+$ denotes the unitary matrix after gate $l$. $O_+=U_+^\dagger OU_+$. The variance of the partial derivative is
> $$Var_{\theta}[\partial_l C]=\int_{U_+}dU_+ \int_{{U_-}}dU_- (\partial_l C(\theta))^2$$
> $$=\int_{{U_+}}dU_+ \int_{{U_-}}dU_- (iTr[U_-\rho U_-^{\dagger} [H_l,O_+]])^2,$$
> where $[\cdot,\cdot]$ denotes the commutator of two matrices.
> $$Var_\theta[\partial_lC]=-\int_{{U_+}}dU_+ (\frac{Tr[\rho^2]Tr[[H_l,O_+]^2]}{d_k^2-1}-\frac{Tr^2[\rho]Tr[[H_l,O_+]^2]}{d_k(d_k^2-1)})$$
> $$=-\int_{U_+}dU_+ (Tr[ [H_l,O_+]^2] \frac{d_k\times Tr[\rho^2]-Tr^2[\rho]}{d_k(d_k^2-1)})$$
> $$=-\frac{d_k\times Tr[\rho^2]-Tr^2[\rho]}{d_k(d_k^2-1)}\int_{{U_+}}dU_+ Tr[ [H_l,O_+]^2] .$$
> The initial state $|\psi_0\rangle$ is set to be the uniform superposition over all computational basis in the $d_k$ subspace. Thus, we have $Tr[\rho]=1,Tr[\rho^2]=1$.
> $$Var_\theta[\partial_lC]=-\frac{1}{d_k(d_k+1)}\int_{\mathcal{U_+}}dU_+ Tr[ [H_l,O_+]^2]$$
> $$=-\frac{2}{d_k(d_k+1)}[Tr(H_lO_+H_lO_+)-Tr(H_lH_lO_+O_+)]$$
> $$=-\frac{2}{d_k(d_k+1)}(\frac{Tr[H^2_l]Tr^2[O]}{d_k^2-1}-\frac{Tr[H^2_l]Tr[O^2]}{d_k(d_k^2-1)}-\frac{Tr[H^2_l]Tr[O^2]}{d_k})$$
> $$=-\frac{2Tr[H_l^2]}{d_k(d_k+1)}(\frac{Tr^2[O]-d_k Tr[O^2]}{d_k^2-1}),$$
> where $Tr[H_l^2]=2*\binom{n-2}{k-1}=\frac{2k(n-k)}{n(n-1)}d_k$. We take $Z_0$ as the observable, then $Tr[O]=\frac{d_k(n-2k)}{n}, Tr[O^2]=d_k$ (other observables will also hold with the same magnitude). Thus, we have
> $$Var_{\theta}[\partial_l C]=-\frac{2}{d_k(d_k+1)}\times\frac{2k(n-k)d_k}{n(n-1)}\times(\frac{\frac{d_k^2(n-2k)^2}{n^2}-d_k^2}{d_k^2-1})$$
> $$=\frac{4k(n-k)}{(d_k+1)n(n-1)}\times\frac{d_k^2(n^2-(n-2k)^2)}{(d_k^2-1)n^2}$$
> $$=\frac{16k^2(n-k)^2 d_k^2}{(d_k+1)n^3(n-1)(d_k^2-1)}\approx \frac{16 k^2(n-k)^2}{n^4 d_k}.$$ $\blacksquare$

---

> > ### Author Response · Authors · 2023-11-16
> >
> > We can further analyze that if the $k$ is only 1, then $Var_\theta[\partial_lC]\approx \frac{16}{n^3}$. If the $k=\frac{n}{2}$ on the other hand, $Var_\theta[\partial_lC]\approx \binom{n}{n/2}^{-1}$, which is approximate to exponentially small. This result is consistent with the conjecture that the trainability of the circuit is closely related to $d_k$ and smaller $d_k$ will lead to better trainability. We will make sure a detailed version of the analysis on the trainability will be included in the revision.
> >
> > **2. We would like to further restate the contributions of our paper and hope this can dissolve your concern.**
> > 1. We revisit the existing works on HW preserving and point out the limitations of these works.
> > 2. We connect various theories to guide the design of HW preserving ansatz. We first utilize the quantum optimal control theory to analyze the capability and expressivity of HW preserving ansatz to see whether the answer to a problem is within the reachable space of the ansatz. We then provide analysis on the trainability of the HW preserving ansatz to see whether we can find a path to reach the answer if it is guaranteed to be in the reachable space by the previous step. Finally, we utilize the overparameterization theory to indicate the approximate number of layers we need to solve the problem.
> > 3. We conducted detailed numerical experiments on constrained VQAs, which are better test cases than [1]. Two tasks involve symmetry-preserving ground state energy estimation (well-studied case) and feature selection (well-studied and investigated by DWave [2]).
> >
> > To the best of our knowledge, we are the first to combine so many theories to provide a thorough analysis of a quantum ansatz. Compared to previous HW preserving papers such as [1], we rectify the theoretical analysis and the utility of the HW preserving ansatz beyond their currently limited applications. Compared to previous works on hard constrained QAOAs [3], we provide a tool for constructing and analyzing the whole group of HW preserving ansatz instead of sieving through existing gates to seek HW preserving properties and use numerical results to decide how to design circuits.
> >
> > To sum up, HW preserving is not a newly proposed idea, but it is never thoroughly analyzed and is often used on unsuitable problems. Some of the theoretical results might have been proposed before but it is novel to combine those theories together for one specific ansatz. We never know why certain HW preserving gates can not solve a given problem and whether the bad results are caused by the fact that the answer is unreachable or poor trainability of the ansatz or lack of layers. We strongly believe that this work can help the QML community by improving the understanding of the HW preserving ansatz. [4] is also a paper that aims to improve the understanding of the community by mostly reviewing the existing ideas and is accepted as a spotlight paper in NeurIPS24. We hope that this work can also be valued and attract more attention to HW preserving circuits.
> >
> >
> > [1] Cherrat, El Amine, et al. "Quantum Deep Hedging." arXiv preprint arXiv:2303.16585 (2023).
> >
> > [2] https://github.com/dwave-examples/feature-selection-notebook/blob/master/01-feature-selection.ipynb
> >
> > [3] Hadfield, Stuart, et al. "From the quantum approximate optimization algorithm to a quantum alternating operator ansatz." Algorithms 12.2 (2019): 34.
> >
> > [4] Abbas, Amira, et al. "On quantum backpropagation, information reuse, and cheating measurement collapse." arXiv preprint arXiv:2305.13362 (2023).

---

> > > ### Author Response · Authors · 2023-11-16
> > >
> > > >***Q1: What are some advantages to directly linking trainability to subspace controllability given the trainability of the HW preserving ansatz was already known?***
> > >
> > > **Ans:** Thanks for your question. We mostly focus on the capability and the expressivity of the HW preserving circuit. Capability and expressivity of a quantum circuit aim at whether we have the answer in the reachable space and the trainability focuses on whether we can reach the answer that is already confirmed in the reachable space. Thus, we believe that the capability and expressivity of quantum ansatz should come along with the discussion about trainability.
> > >
> > > For example, if we only have the results from [1] (which we suspect might not be correct) and we are trying to solve the ground state energy estimation problem with RBS gate on the nearest neighbor physical qubit connected quantum hardware. It is impossible for us to get the answer according results in our paper, which is because the answer is not even in the reachable subspace of Nearest-Neighbor connected RBS gates. However, [1] shows that RBS gate is trainable and should be able to get a reasonable answer. So only analyzing the trainability of an ansatz is far from enough to understand the characters of an ansatz.
> > >
> > > Moreover, we have encountered situations in which some colleagues of ours failed to get reasonable answers even with both the capability and trainability of the ansatz confirmed. It turns out that they do not concatenate enough layers to reach the results. That is why we provide results regarding the overparameterization theory to analyze how many HW preserving gates we need to reach a reasonable answer.
> > >
> > > We have also stated in the paper that using HW preserving circuits on quantum kernels is a waste of the talent of HW preserving circuits and we strongly believe the battlefield of HW preserving circuits should be in the constrained VQAs and that's why we propose this work.
> > >
> > > We hope that we have dissolved your concerns and we are eager to hear back from you.
> > >
> > >
> > > [1] Cherrat, El Amine, et al. "Quantum Deep Hedging." arXiv preprint arXiv:2303.16585 (2023).

---

> ### Comment · Reviewer_obpr · 2023-11-16
>
> I believe the authors are referring to the final Equation in Section 4 of arXiv:2303.16585, that is, the final equation in the proof of Theorem 2. In this proof the authors of arXiv:2303.16585 are interested only in proving an asymptotic lower-bound of $\frac{1}{n^6}$---this lower-bound agrees with yours asymptotically as $\frac{k(n-k)}{n(n-1)}$ is asymptotically larger than $\frac{1}{n^6}$, so I am unclear on the discrepancy.
>
> To address the authors' other comments: I unfortunately still do not believe there is enough novelty in this work. As the authors mention, they do indeed perform additional numerical tests of the HW preserving ansatz---I have upped my contribution score accordingly---but in my opinion this is not enough to warrant acceptance. Outside of trainability, why is the HW preserving ansatz better suited for these kinds of problems rather than the finance problems considered in arXiv:2303.16585? In my opinion this essentially is just more of the same kind of numerics already previously run, just in a slightly different context.

---

> > ### Author Response · Authors · 2023-11-17
> >
> > We are grateful for your in-time response and we are happy to further discuss with you.
> >
> > >***For the discrepancy of the trainability calculation.***
> >
> > Firstly, we would like to be more clear about the difference between our calculation and the results in [1]. **We provide a completely different bound on the trainability** (comparison shown in the following table). We derivate a whole new bound by integrating all the unitary matrices before and after the $l$-th layer instead of taking the parameters as 0 (as [1] did in their paper). We provide the whole proof and derivation of the bound in the previous reply.
> >
> > |          | $Var_\theta[\partial_lC]$ | $Var$ with $k=1$ | $Var$ with $k=n/2$
> > | -------- | -------- | -------- |  -------- |
> > | Ours     | $\frac{16k^2 (n-k)^2}{n^4 d_k}$ | $\frac{16}{n^3}$  | $\tbinom{n}{n/2}^{-1}$ |
> > | [1]      | $4(\frac{k(n-k)}{n(n-1)})^4$ | $\frac{1}{n^4}$ | constant number |
> >
> > The $Var_\theta[\partial_lC]=\frac{1}{2}(\partial_lC_k)^2$ as shown in Theorem 1 in [1].
> >
> > Secondly, as for the results in [1]. For the last equation in theorem 2, we have
> > $$\partial_lC_k=\langle\psi_k|(\sigma_X^i\otimes\sigma_X^j+\sigma_Y^i\otimes\sigma_Y^j)|\psi_k\rangle=2(\frac{2\tbinom{n-2}{k-1}}{\tbinom{n}{k}})^2=\Omega(\frac{1}{n^6}).$$
> > We have the following three reasons to say why it seems incorrect.
> > 1. If $\partial_lC_k =8(\frac{k(n-k)}{n(n-1)})^2$ is correct in [1], then the usage of Omega notation is incorrect. Big omega notation indicate a larger and equal to, but we don't see under any circumstances that $8(\frac{k(n-k)}{n(n-1)})^2$ is equal to $\frac{1}{n^6}$. $$\partial_lC_k=\langle\psi_k|(\sigma_X^i\otimes\sigma_X^j+\sigma_Y^i\otimes\sigma_Y^j)|\psi_k\rangle=2(\frac{2\tbinom{n-2}{k-1}}{\tbinom{n}{k}})^2\neq\Omega(\frac{1}{n^6})$$
> > 2. According to our calculation **following the scheme in [1]**, $\partial_lC_k$ should be $4\frac{k(n-k)}{n(n-1)}$ instead of $8(\frac{k(n-k)}{n(n-1)})^2$. This is a clear error. $$\partial_lC_k=\langle\psi_k|(\sigma_X^i\otimes\sigma_X^j+\sigma_Y^i\otimes\sigma_Y^j)|\psi_k\rangle\neq2(\frac{2\tbinom{n-2}{k-1}}{\tbinom{n}{k}})^2\neq\Omega(\frac{1}{n^6})$$
> > 3. When $k=n/2$, the bound is a constant number without any $n$ in it is clearly incorrect. $$\partial_lC_k\neq\langle\psi_k|(\sigma_X^i\otimes\sigma_X^j+\sigma_Y^i\otimes\sigma_Y^j)|\psi_k\rangle\neq2(\frac{2\tbinom{n-2}{k-1}}{\tbinom{n}{k}})^2\neq\Omega(\frac{1}{n^6})$$
> >
> > Thus, we conclude that the results in [1] is incorrect and therefore we provide our own bound based on a completely different scheme.

---

> > > ### Author Response · Authors · 2023-11-17
> > >
> > > >***For the comparison between the experiment tasks in our work and in [1]***
> > >
> > > We would like to first clarify the difference between quantum deep neural networks (QDNN) and variational quantum algorithms (VQA). Quantum deep neural networks focus on designing a quantum kernel analog to classical DNN. In most cases, it is supervised learning with a training set to train a quantum circuit as a model and make predictions on the test set. Variational quantum algorithms, especially variational quantum eigensolver, on the other hand, evolve a quantum state to a ground state with minimum energy. They do not have training or test sets, and QDNN and VQA are two completely different quantum machine learning tasks.
> > >
> > > Our claim in the work is that **HW preserving circuit is more suitable for VQAs instead of QDNNs**. We have the claim for the following reasons:
> > > 1. All the good properties such as trainability come from the limited subspace. We need to make sure all the training and testing sets are in the subspace. If the initial states are not in the HW preserving subspace, then it is wasted.
> > > 2. We need to make sure only the HW preserving layers are used since once other layers that are not HW preserving are involved, we can never restrict the space back to the HW preserving subspace. QDNN usually stacks different layers such as the quantum convolution layer, and quantum pooling layer to help improve the results that do not preserve the HW.
> > > 3. The pipeline of VQA is to set an initial state in a space, and then evolve the state in the space to find the best state as the output. Once we ensure the initial state is in the subspace and all the gates we use preserve the state in the subspace, then we are able to find the best state in this very subspace.
> > >
> > > We can provide a simple example to further illustrate why VQA is more suitable. Say we have a $n=4$ and $k=2$ subspace. Then only 6 out of 16 states are in the subspace, which are $0011, 1100,1001,0110,1010,0101$. If we operate QDNN in this subspace, we need to make sure all the training and testing set is within the linear combination of the 6 states (it is very hard to find such a dataset with actual meaning). We also need to ensure not a single gate that will produce an illegal state such as 1110 is allowed in the circuit (including H, X, Y, Z, CNOT, CZ). That is why the HW preserving circuit is not very suitable for QDNN problems. However, for a VQA problem, we only need to take any of the 6 possible states as the initial state and use RBS gate (as an example) to evolve the quantum system to get the lowest energy. The energy should be a special combination of the 6 possible states and the whole process will be in the $n=4,k=2$ subspace. Therefore, we conclude that VQA is more suitable for HW preserving circuits than QDNN.
> > >
> > > As for why our experiments exceed those in [1]:
> > > 1. Quantum chemistry problem requires extreme accuracy where we can see that we reach an error smaller than $1e-12$, which is very hard to reach. Finance problems usually do not even have the ground truth e.g. when we are predicting the stock price.
> > > 2. We provide stronger baselines including both SOTA methods and well-studied famous methods. [1] does not compare to existing quantum or classical methods to demonstrate their supremacy.
> > > 3. Quantum chemistry problems (or more specifically the ground state energy estimation problem) have been studied by the quantum computing community for a long time and there are numerous papers published on both science and nature regarding this area ([2,3,4], HEA used in [4] is one of the baselines in our work).
> > >
> > >
> > > To sum up, QDNN and VQA tasks are completely different tasks and VQA is a better fit for HW preserving circuits. We also present way stronger results than [1]. We hope that we can dissolve your concerns and we would be grateful maybe you could consider raising the overall rating of our work.
> > >
> > >
> > > [1] [1] Cherrat, El Amine, et al. "Quantum Deep Hedging." arXiv preprint arXiv:2303.16585 (2023).
> > >
> > > [2] Lanyon, Benjamin P., et al. "Towards quantum chemistry on a quantum computer." Nature Chemistry 2.2 (2010): 106-111.
> > >
> > > [3] Yuan, Xiao. "A quantum-computing advantage for chemistry." Science 369.6507 (2020): 1054-1055.
> > >
> > > [4] Kandala, Abhinav, et al. "Hardware-efficient variational quantum eigensolver for small molecules and quantum magnets." nature 549.7671 (2017): 242-246.

---

> > > > ### Author Response · Authors · 2023-11-19
> > > >
> > > > Dear reviewer obpr,
> > > >
> > > > As the discussion period is approaching the deadline, we are anxious to know if we have solved all your worries. If you have any further questions please get in touch with us and we will try our best to clarify them.
> > > >
> > > > Sincerely
> > > >
> > > > #3179 Authors

---

> > > > > ### Comment · Reviewer_obpr · 2023-11-21
> > > > >
> > > > > I believe the authors are misunderstanding the typical use of $\Omega$ notation---just because the bound is not saturated (i.e., it is loose) does not mean it is not true. For instance, $n^2$ is $O(n^3)$, even though it is also $o(n^3)$. It just shows that the bound is not tight. In the case of arXiv:2303.16585 this is fine since the goal is just to show that the gradients are at least $\sim\frac{1}{\operatorname{poly}(n)}$.
> > > > >
> > > > > I am also unclear on the authors' argument that arXiv:2303.16585 fails when the input does not have definite Hamming weight, as this requirement is stated in Theorem 2 of arXiv:2303.16585.
> > > > >
> > > > > Accordingly, though I agree with the authors that there are slight improvements to previously stated results, I unfortunately still disagree with the impact of the work.

---

> ### Author Response · Authors · 2023-11-22
>
> We would like to set the arguments regarding [1] apart since we are reviewing our work instead of [1] here, the correctness of [1] is not an issue that we should focus on. We do acknowledge that [1] is a precious work for quantum algorthms in finance problems with all the materials provided. We cite this paper with full respect and it helps us with a better understanding of the HW preserving ansatz. The improvements of our submission compared to previous works (not limited to [1]) are listed as follows:
>
> 1. We thoroughly analyze the HW preserving ansatz by utilizing theoretical tools on the capability, expressivity, and trainability (with a tighter bound on the trainability than previous works).
> 2. We conduct experiments on the constrained VQA problem which is well-studied and preferred by the quantum computing society and we have achieved much better and steady results.
> 3. We are the first to only use HW preserving gates to accomplish the unitary approximation task indicating that a single HW preserving gate can be universal in the HW preserving subspace.
> 4. We also state that parity check for bit flip error is much easier on the HW preserving ansatz, which shows the potential ability of HW preserving ansatz in the NISQ era.
>
> Therefore, we do believe that we have some improvements to the previous works. Considering the fact that you agree with the soundness (4) and presentation (3) of our work, we believe our work should be $\Omega(5)$ in the overall rating according to your assessment. We truly hope the reviewer to “Be mindful of potential biases and try to be open-minded about the value and interest a paper can hold for the entire ICLR community, even if it may not be very interesting for you.” (in the reviewer guide "reviewing a submission: step-by-step" point 2.4)
>
>  [1] Cherrat, El Amine, et al. "Quantum Deep Hedging." arXiv preprint arXiv:2303.16585 (2023).

---

### Official Review · Reviewer_Yeih · 2023-10-27

**Soundness:** 4 excellent
**Presentation:** 3 good
**Contribution:** 2 fair
**Rating:** 6
**Confidence:** 4

**Summary:**

This paper studies ansatz for variational quantum algorithms with the requirement that they preserve Hamming weight. They test the expressibility for the task of unitary approximation, where they show that the expressibility of these ansatz is the same as predicted (in terms of the number of parameters needed). In addition, experiments demonstrate the effectiveness of their approach on two different tasks: ground state preparation and feature selection.

**Strengths:**

- This paper nicely summarizes and makes clear the idea of using Hamming weight (or in physics terminology, excitation) preserving ansatz for variational quantum algorithms. The numerical results verify the accuracy of the theory, which characterizes the expressibility of these ansatz.
- The experiments are nice, showing a modest improvement over existing methods.

**Weaknesses:**

- Typos: "relative" should be "relatively" throughout
- Typo: In Figure 2, "Haar measurement" should be "Haar measure"
- Typo: "Hartree-Fork"
- Based on common gate sets of real machines, I feel the BS gate is quite artificial. It might make more sense to use parameterized $XX$ rotations with single-qubit $Z$ rotations. If the intention is to perform VQAs on near-term machines, then one might want to use more native instructions than the decompositions shown in Fig 5 and Fig 6.
- Related to the above point, the decomposition of the BS gate is addressed as a possible limitation. I am pretty sure a better decomposition is possible using $XX$, $YY$ and $ZZ$ rotations.
- I think this idea of preserving the Hamming weight is not very new, and the theoretical results are not very surprising to me. For this reason, I feel the contribution of this work is not so large, although the proposed ansatz appear to perform better than existing ones.

**Questions:**

- In Equation 8, should the maximization be over $M$ such that $M\geq M_c$?
- Where does the construction for the BS gate come from?
- Throughout this paper, it is stated that this is a "revisiting" of the Hamming weight preserving ansatz. What is the status of the previous work using these ansatz, and what are the specific improvements made in this paper compared to previous proposals?

---

> ### Author Response · Authors · 2023-11-16
>
> >***W1: Based on common gate sets of real machines, I feel the BS gate is quite artificial. It might make more sense to use parameterized XX rotations with single-qubit Z rotations. If the intention is to perform VQAs on near-term machines, then one might want to use more native instructions than the decompositions shown in Fig 5 and Fig 6.***
>
> **Ans:** Thanks for your valuable suggestion. We do acknowledge that the BS gate is quite 'artificial' since we are trying to construct a single gate that can be universal with both Nearest-Neighbor and Full-Connect connectivity. One of the key factors for HW preserving gate is that the Hamiltonian for the two-qubit parameterized gate is $$H_{HW}=\left(\begin{array}{cccc}
>         0 & 0 & 0 & 0\\\        0 & a & b & 0\\\        0 & b^\dagger & c & 0\\\        0 & 0 & 0 & 0
>     \end{array}\right).$$
>
> It is still quite hard to construct a legit Hamiltonian using XX, YY, and ZZ rotations since only ZZ rotations are HW preserving.
>
> We could further use the combination of XX, YY, and ZZ rotations to construct HW preserving gates. As mentioned by Reviewer ADjN, XY-mixer is another well-studied HW preserving gate. The hamiltonian of XY-mixer is $H_{XY}=X\otimes X+Y\otimes Y$, and the unitary matrix of XY-mixer is
> $$U_{XY}=\left(\begin{array}{cccc}
>         1 & 0 & 0 & 0\\\        0 & \cos (\theta) & -\text{i}\sin (\theta) & 0\\\        0 & -\text{i}\sin (\theta) & \cos (\theta) & 0\\\        0 & 0 & 0 & 1
>     \end{array}\right).$$
>
> We would like to point out that **the decomposed circuit of those HW preserving gates constructed with XX, YY, and ZZ is not very simple.** Take XY-mixer as an example. Although using XX+YY to construct the Hamiltonian is very elegant, the unitary matrix of the XY-mixer is not very easy to decompose (even if we have iSWAP gate on superconducting hardware). **RBS gate, on the other hand, can be a built-in gate on photonic quantum computers.** Therefore, those that are difficult to decompose on superconducting hardware might be very easy to fulfill in other quantum systems.
>
> As we have stated right before section 5, despite the fact that RBS and XY-mixer do not have full DLA dimension, if we pick the right problem, they can also be better choices than BS gate. We could also use the combination of RBS and RZZ gates to surpass BS-gate-only circuits. BS gate is only an example to illustrate how to construct a single HW preserving gate with full DLA even under harsh conditions (nearest neighbor connectivity for physical qubits)
>
> >***W2: Related to the above point, the decomposition of the BS gate is addressed as a possible limitation. I am pretty sure a better decomposition is possible using XX, YY, and ZZ rotations.***
>
> **Ans:** Thanks for your thoughtful advice. We are seeking a better decomposition of the BS gate to make it more hardware-friendly. However, it seems hard to use XX, YY, and ZZ to decompose the Hamiltonian of the BS gate. The Hamiltonians for XX, YY, and ZZ are
> $$H_{XX}=\left(\begin{array}{cccc}
>         0 & 0 & 0 & 1\\\        0 & 0 & 1 & 0\\\        0 & 1 & 0 & 0\\\        1 & 0 & 0 & 0
>     \end{array}\right),\ H_{YY}=\left(\begin{array}{cccc}
>         0 & 0 & 0 & -1\\\        0 & 0 & 1 & 0\\\        0 & 1 & 0 & 0\\\        -1 & 0 & 0 & 0
>     \end{array}\right),\ H_{ZZ}=\left(\begin{array}{cccc}
>         1 & 0 & 0 & 0\\\        0 & -1 & 0 & 0\\\        0 & 0 & -1 & 0\\\        0 & 0 & 0 & 1
>     \end{array}\right),\ $$
>
> It seems that we can only construct HW preserving gates with $a=c$ and $b=b^\dagger$. These gates such as XY-mixer can not reach full DLA dimension with Nearest-Neighbor connectivity. There is a chance that we misunderstood the weakness brought up by the reviewer so we are eager to hear back from you.

---

> > ### Author Response · Authors · 2023-11-16
> >
> > >***W3: I think this idea of preserving the Hamming weight is not very new, and the theoretical results are not very surprising to me. For this reason, I feel the contribution of this work is not so large, although the proposed ansatz appear to perform better than existing ones.***
> >
> > >***Q3: Throughout the paper, it is stated that this is a "revisiting" of the Hamming weight preserving ansatz. What is the status of the previous work using these ansatz, and what are the specific improvements made in this paper compared to previous proposals?***
> >
> > **Ans:** Thanks for your comments. We will answer the above two questions and weaknesses together.
> >
> > As suggested by Reviewer ADjN, we have provided an analysis on the trainability of HW preserving ansatz (see the answer for weakness 3 for ADjN for detailed derivation). We initially omit the analysis on the trainability since [1] has provided one derivation. However, after careful review, we believe the results in [1] are incorrect and thus we provide our own derivation. We would like to restate the contributions of this work:
> > 1. We revisit the existing works on HW preserving and point out the limitations of these works.
> > 2. We connect various theories to guide the design of HW preserving ansatz. We first utilize the quantum optimal control theory to analyze the capability and expressivity of HW preserving ansatz to see whether the answer to a problem is within the reachable space of the ansatz. We then provide analysis on the trainability of the HW preserving ansatz to see whether we can find a path to reach the answer if it is guaranteed to be in the reachable space by the previous step. Finally, we utilize the overparameterization theory to indicate the approximate number of layers we need to solve the problem.
> > 3. We conducted detailed numerical experiments on constrained VQAs, which are better test cases than [1]. Two tasks involve symmetry-preserving ground state energy estimation (well-studied case) and feature selection (well-studied and interested by DWave [2]).
> >
> > To the best of our knowledge, we are the first to combine so many theories to provide a thorough analysis of a quantum ansatz. Compared to previous HW preserving papers such as [1], we rectify the theoretical analysis and the utility of the HW preserving ansatz beyond their currently limited applications. Compared to previous works on hard constrained QAOAs [3], we provide a tool for constructing and analyzing the whole group of HW preserving ansatz instead of sieving through existing gates to seek HW preserving properties and use numerical results to decide how to design circuits.
> >
> > To sum up, HW preserving is not new, but it is never thoroughly analyzed and is often used on unsuitable problems. The theoretical results might not be surprising to you but we never know why certain gates can not solve a given problem and whether the bad results are caused by the fact that the answer is unreachable or poor trainability of the ansatz or lack of layers. We strongly believe that this work can help the QML community by improving the understanding of the HW preserving ansatz. [4] is also a paper that aims to improve the understanding of the community by mostly reviewing the ideas and is accepted as a spotlight paper in NeurIPS24. We hope that this work can also be valued and attract more attention to HW preserving circuits.
> >
> >
> > [1] Cherrat, El Amine, et al. "Quantum Deep Hedging." arXiv preprint arXiv:2303.16585 (2023).
> >
> > [2] https://github.com/dwave-examples/feature-selection-notebook/blob/master/01-feature-selection.ipynb
> >
> > [3] Hadfield, Stuart, et al. "From the quantum approximate optimization algorithm to a quantum alternating operator ansatz." Algorithms 12.2 (2019): 34.
> >
> > [4] Abbas, Amira, et al. "On quantum backpropagation, information reuse, and cheating measurement collapse." arXiv preprint arXiv:2305.13362 (2023).
> >
> > >***W4: Some typos in the paper***
> >
> > **Ans:** Thanks for your careful review and we will correct these typos in the revision.

---

> > > ### Author Response · Authors · 2023-11-16
> > >
> > > >***Q1: In Equation 8, should the maximization be over $M$ such that $M\geq M_c$?***
> > >
> > > **Ans:** We apologize for the mistake. We will correct this in the revision.
> > >
> > > >***Q2: Where does the construction for the BS gate come from?***
> > >
> > > **Ans:** Thanks for your comments. The construction for the BS gate comes from the general Hamiltonian form of the HW preserving gate, which is
> > > $$
> > > H_{HW}=\left(\begin{array}{cccc}
> > >         0 & 0 & 0 & 0\\\        0 & a & b & 0\\\        0 & b^\dagger & c & 0\\\        0 & 0 & 0 & 0
> > >     \end{array}\right).
> > > $$
> > > As stated in the paper, we can construct an arbitrary HW preserving gate by deciding the values for $a,b,c$. BS gate is only an example to illustrate a gate with full DLA dimension with both NN and FC connectivity. Considering the fact that $H_{HW}$ is a Hermitian matrix, we have the following situations. For $a$ and $c$, we have $a=c$ or $a\neq c$. For $b$, we have $b=b^\dagger$ where $b$ is a real number or $b=-b^\dagger$ where $b$ only has the imaginary part or $b$ is an arbitrary imaginary number with both real and imaginary parts. We have tried different combinations of the $a,b,c$ and come up with the BS gate to be universal for both connectivity (where $a=c\neq 0$ and $b$ has both imaginary and real parts).
> > >
> > >
> > > As commented by Reviewer ADjN, we provide "a bottom-up approach to guide quantum circuit design compared to previous top-down approach." Actually, the existing HW preserving gates, such as RBS gate and XY-mixer, are top-down approaches, where they first come up with a gate (maybe with $H=X\otimes X+Y\otimes Y$) and then explore the properties of whether it is HW preserving. It is just like mining in a huge gate mountain to find some golden gates that are HW preserving. Therefore, we provide a way to theoretically construct an HW preserving gate so that we can have enough samples to have a deeper understanding regarding the HW preserving ansatz.

---

> > > > ### Comment · Reviewer_Yeih · 2023-11-19
> > > >
> > > > Thanks for the detailed responses. I'm fairly satisfied with the answers and I'll keep my score. Regarding the decomposition of the BS gate, I was mistaken when I said it could be decomposed using (only) XX, YY, and ZZ rotations, though I am still certain that better decompositions are possible depending on the native operations allowed by the hardware (which are often Pauli rotations such as XX rotations).

---

> > > > > ### Author Response · Authors · 2023-11-21
> > > > >
> > > > > Thanks for your continuous effort in reviewing our paper and your valuable suggestions. We will further investigate the decomposition of the HW preserving gate to make it truly useful in the NISQ era.
> > > > >
> > > > > best

---

### Official Review · Reviewer_Eu68 · 2023-11-01

**Soundness:** 4 excellent
**Presentation:** 4 excellent
**Contribution:** 3 good
**Rating:** 8
**Confidence:** 3

**Summary:**

Different families of Hamming weight (HW) preserving quantum circuits are examined in the context of general variational algorithms. A theoretical framework is developed for understanding when a given circuit family is sufficiently expressive to generate arbitrary symmetric transformations, and these various circuit families are experimentally compared against each other and against different baseline models in a variety of problems.

**Strengths:**

* The paper includes a large number of results of both a theoretical and experimental nature, that should be useful for someone wanting to use HW-preserving quantum circuits. Overall, this will likely encourage the use of these type of symmetric circuits in the context of more general classes of variational problems than they have been used in the past.

* The experimental tasks are well chosen, with the unitary approximation, ground state energy estimation, and feature selection problems being representative of problems from quantum computing, quantum chemistry, and machine learning. The baseline methods used for comparison seem to be well chosen, and to the best of my knowledge are representative of the methods actually used in these various subject areas.

* The paper is well-written, both in terms of its overall structure and its writing. Given the amount of material presented, this is critical for being able to understand the paper's results. The appendices are quite helpful for providing background and additional detail about the results.

**Weaknesses:**

* The premise of the paper is a bit atypical for conference submissions, in that it isn't "selling" a new model or method developed by the authors. Rather, the goal of the paper seems to be improving the community's understanding regarding the expressivity and performance of HW-preserving circuits in general. I think this is a valuable contribution, but I could see other reviewers pointing to a lack of novelty due to this unorthodox aim.

* There is a lot of material that is packed into a limited space, and while I found the presentation to help a lot with this, the results still take some time to digest and understand in detail.

* Figure 4 compares a large number of different circuit ansatzes, but many of these overlap with each other and are barely visible (e.g. the RBS-full points). It would helpful to revise this figure to permit easier viewing of all of the baseline results, for example by varying the colors, size, and/or ordering of the plot markers.

**Questions:**

I don't have any particular questions for the authors.

---

> ### Author Response · Authors · 2023-11-16
>
> >***W1: The premise of the paper is a bit atypical for conference submissions, in that it isn't "selling" a new model or method developed by the authors. Rather, the goal of the paper seems to be improving the community's understanding regarding the expressivity and performance of HW-preserving circuits in general. I think this is a valuable contribution, but I could see other reviewers pointing to a lack of novelty due to this unorthodox aim.***
>
> **Ans:** We are grateful for your appreciation of our work. We truly hope that this work can be valued and attract more attention to HW preserving ansatz.
>
>
> >***W2: There is a lot of material that is packed into a limited space, and while I found the presentation to help a lot with this, the results still take some time to digest and understand in detail.***
>
> **Ans:** According to the suggestion from Reviewer ADjN, we have added the analysis on the trainability of HW preserving ansatz. There will be more detailed material in this paper. We will try our best to be clear and concise on the key points and provide enough theoretical backups as well.
>
> >***W3: Figure 4 compares a large number of different circuit ansatzes, but many of these overlap with each other and are barely visible (e.g. the RBS-full points). It would be helpful to revise this figure to permit easier viewing of all of the baseline results, for example by varying the colors, size, and/or ordering of the plot markers.***
>
> **Ans:** Thanks for your valuable advice and we have adjusted Figure 4 according to your suggestion. The revised figure is shown in https://anonymous.4open.science/r/ICLR3179/SupplementaryMaterial.pdf.

---

### Official Review · Reviewer_ADjN · 2023-11-01

**Soundness:** 3 good
**Presentation:** 2 fair
**Contribution:** 3 good
**Rating:** 8
**Confidence:** 5

**Summary:**

This work discusses incorporating symmetries as hard constraints in ansatz design for Variational
Quantum Algorithms (VQAs). In particular, the authors revisit the Hamming Weight (HW) preserving ansatz. By adopting Dynamical Lie Algebra (DLA) as a quantum optimal control theory and over-parameterization theory based on the maximum rank of quantum Fisher information matrix, one can quantify (and further ensure) the expressivity and capability of the proposed symmetry-preserving ansatz. These theoretical guarantees are verified on the unitary approximation problem. Moreover, the authors can demonstrate better performance on ground-state energy estimation ( need to preserve electron number) and feature selection as a constrained QUBO problem (preserve number of features), both using VQE.

**Strengths:**

A novel idea to use Dynamical Lie Algebra and over-parameterization theory for Parametrized Quantum Circuit (PQC) to design symmetry-preserving VQA ansatz.
Very clear explanation of the theoretical tools used.
The authors conduct extensive numerical studies showing positive results.
A bottom-up approach to guide quantum circuit design compared to the previous top-down approach [1]. The latter uses a different sampling-based definition of expressivity and entanglement capability to guide general-purpose circuit design, but actual task performance could vary a lot.
Hard constraints are more favorable than soft constraints in some industrial use cases, where solution validity and robustness are priorities.
[1] Sim, Sukin, Peter D. Johnson, and Alán Aspuru‐Guzik. "Expressibility and entangling capability of parameterized quantum circuits for hybrid quantum‐classical algorithms." *Advanced Quantum Technologies* 2.12 (2019): 1900070.

**Weaknesses:**

Regarding the literature review, it is not clear to me why the authors chose to categorize some previous work based on XY-mixer QAOA [1-2] as soft constraints. Their circuit is composed of a problem Hamiltonian layer (ZZ gates) and a mixer layer, both preserving the Hamming weight instead of adding a penalty to the cost function.
One of my biggest concerns is when we run this framework on NISQ hardware or noisy simulation. A bit-flip (X) error would suddenly break the Hamming distance. Will this be a big problem or not? It will be better to see more results along this consideration.
Need more discussion on the trainability of the proposed ansatz and Barren Plateau (BP) phenomenon. It makes intuitive sense to think of better trainability compared to HEA since it is only exploring a constrained subspace. The optimization problem should be easier. Previous work [3] showed that a compound layer consisting of FBS gates could lead to a gradient decay only polynomially. Will the same conclusion hold for this work?
Some numerical experiment details don't make sense to me. For example, why did the authors choose the penalty weight in Eq. 16 as \alpha = [0.5, 1, 5, 10]. To my best knowledge, people usually set this empirical parameter to a much larger value like 100.

1] Hadfield, Stuart, et al. "From the quantum approximate optimization algorithm to a quantum alternating operator ansatz." *Algorithms* 12.2 (2019): 34.
[2] He, Zichang, et al. "Alignment between initial state and mixer improves qaoa performance for constrained portfolio optimization." *arXiv preprint arXiv:2305.03857* (2023).
[3] Cherrat, El Amine, et al. "Quantum Deep Hedging." *arXiv preprint arXiv:2303.16585* (2023).

**Questions:**

Can you explain why you use gate infidelity as the performance metric for the unitary approximation problem but plot Success Probability in Fig. 2? I guess the authors are converting infidelity of 1e-5 as log10(1e-5)/log10(1e-10)=0.5. Please make it clear in the paper.
It would be nice if the author could discuss connections of their HW preserving VQE with XY-mixer QAOA or other constrained QAOA. And maybe apply the theoretical tools used here to improve those work?
What is the basis set used in ground-state energy estimation (such as STO-3G) to discretize the computation space?
I believe the second test case, feature selection, is not a typical benchmark for VQAs. Is this a classically hard problem? Can you say more about why choose this test case?

Some Typos:
--Right below Lemma 4.1. "Ramakrishna et al. (1995) has shown that if the dimension of DLA" instead of "Ramakrishna et al. (1995) has shown that if the dimension of the dimension of DLA"
The y-axis in Fig. 3 and Fig. 7 should be in logarithmic scale
Right below Appendix B.1. 'qubit'

---

> ### Author Response · Authors · 2023-11-16
>
> >***W1: Why the authors chose to categorize some previous work based on XY-mixer QAOA as soft constraints***
>
> **Ans:** We apologize for the misleading description of related works. We cite [1] to illustrate that there are quite a lot of constrained QAOA problems and then we claim that soft constraints could be a solution to these constrained optimization problems. The citation is very close to the following claim but we do not intentionally say the models in [1] are soft-constrained. We do confirm that XY-mixer QAOA is a hard-constrained QAOA, and we will make a clear intro on soft-constrained and hard-constrained models in related works in the revision.
>
> [1] Hadfield, Stuart, et al. "From the quantum approximate optimization algorithm to a quantum alternating operator ansatz." Algorithms 12.2 (2019): 34.
>
>
> >***W2: Bit-flip error on NISQ hardware will be a big problem for HW preserving circuits.***
>
> **Ans:** Thanks for bringing up this problem which is crucial to improving our work. It is quite interesting that **HW preserving circuit is much easier to be bit-flip fault tolerant**.
> We can utilize one ancilla qubit to do the parity check to detect any bit-flip error on the HW preserving circuit. The Hamming weight $k$ implies whether we have an odd number or even number of $1$s in the circuit. One possible solution of the parity check is to link all the working qubits to an ancilla qubit with CNOT gate with the ancilla qubit as the target bit. We initially set the ancilla qubit to $|0\rangle$. If we have an odd number of $1$s in the working qubits, we will have a $|1\rangle$ on the ancilla qubit and a $|0\rangle$ with an even number of $1$s. Thus, it is much easier to detect a bit-flip error on the HW preserving circuit compared to Shor's code with two ancilla qubits required to protect one qubit from bit-flip error. We can either rerun the circuit or locate the qubit to correct the bit-flip error. This is an exciting and primitive conclusion, and we hope this dissolves your concern.

---

> > ### Author Response · Authors · 2023-11-16
> >
> > >***W3: More discussion on the trainability is required.***
> >
> > **Ans:** Thanks for your thoughtful advice. We carefully reviewed the results and proof in [1] after seeing your comments and we are surprised to find it incorrect. There are still chances that we misunderstood the calculation for the variance bound in [1], so we decided to give our results here with a sketch proof. Detailed derivation will be included in the revision.
> >
> > We first discuss the results shown in [1]. We tried the derivation of theorem 2 and find that $\partial_lC_k=\langle\psi_0|(\sigma_X^i\otimes\sigma_X^j+\sigma_Y^i\otimes\sigma_Y^j)|\psi_0\rangle$ should be $1$ instead of $\frac{1}{4}$. For the initial state in the HW k subspace, the results in [1] are $\partial_lC_k=\langle\psi_k|(\sigma_X^i\otimes\sigma_X^j+\sigma_Y^i\otimes\sigma_Y^j)|\psi_k\rangle=2(\frac{2\tbinom{n-2}{k-1}}{\tbinom{n}{k}})^2=\Omega(\frac{1}{n^6})$, where according to our derivation, the results should be $\partial_lC_k=\langle\psi_k|(\sigma_X^i\otimes\sigma_X^j+\sigma_Y^i\otimes\sigma_Y^j)|\psi_k\rangle=2\frac{2\tbinom{n-2}{k-1}}{\tbinom{n}{k}}$, $\frac{\tbinom{n-2}{k-1}}{\tbinom{n}{k}}=\frac{k(n-k)}{n(n-1)}$. Even if the original result is correct that $\partial_lC_k=8(\frac{k(n-k)}{n(n-1)})^2$, $\partial_lC_k=\mathcal{O}(1)$ if we take $k=n/2$, which seems incorrect.
> >
> >
> > Now we provide the results based on our derivation. Without loss of generality, we take RBS gate as an example since [1] also derives the bound based on RBS gate.
> >
> > ***Theorem 1. Consider a quantum circuit operating in the subspace with Hamming Weight $k$. The variance of the cost function partial derivative is $$Var_\theta[\partial_lC]\approx \frac{16k^2(n-k)^2}{n^4d_k}$$***
> >
> > *Proof.* Consider the partial derivative of the cost function $C$ with respect to the parameters $\theta$. For some parameter $\theta_l$ in the $l$-th RBS gate, we have:
> > $$\partial_{l} C(\theta)=\partial_l(Tr[U(\theta)\rho U(\theta)^{\dagger}O])$$
> > $$=\partial_l(Tr[U_-\rho U_-^{\dagger} O_+]) $$
> > $$=iTr[U_-\rho U_-^{\dagger} [H_l,O_+]],$$
> >
> > where $\rho$ is the input state, $O$ is the observable to measure, $U_-$ denotes the unitary matrix of the circuit before the $l$-th gate and $U_+$ denotes the unitary matrix after gate $l$. $O_+=U_+^\dagger OU_+$. The variance of the partial derivative is
> > $$Var_{\theta}[\partial_l C]=\int_{U_+}dU_+ \int_{{U_-}}dU_- (\partial_l C(\theta))^2$$
> > $$=\int_{{U_+}}dU_+ \int_{{U_-}}dU_- (iTr[U_-\rho U_-^{\dagger} [H_l,O_+]])^2,$$
> > where $[\cdot,\cdot]$ denotes the commutator of two matrices.
> > $$Var_\theta[\partial_lC]=-\int_{{U_+}}dU_+ (\frac{Tr[\rho^2]Tr[[H_l,O_+]^2]}{d_k^2-1}-\frac{Tr^2[\rho]Tr[[H_l,O_+]^2]}{d_k(d_k^2-1)})$$
> > $$=-\int_{U_+}dU_+ (Tr[ [H_l,O_+]^2] \frac{d_k\times Tr[\rho^2]-Tr^2[\rho]}{d_k(d_k^2-1)})$$
> > $$=-\frac{d_k\times Tr[\rho^2]-Tr^2[\rho]}{d_k(d_k^2-1)}\int_{{U_+}}dU_+ Tr[ [H_l,O_+]^2] .$$
> > The initial state $|\psi_0\rangle$ is set to be the uniform superposition over all computational basis in the $d_k$ subspace. Thus, we have $Tr[\rho]=1,Tr[\rho^2]=1$.
> > $$Var_\theta[\partial_lC]=-\frac{1}{d_k(d_k+1)}\int_{\mathcal{U_+}}dU_+ Tr[ [H_l,O_+]^2]$$
> > $$=-\frac{2}{d_k(d_k+1)}[Tr(H_lO_+H_lO_+)-Tr(H_lH_lO_+O_+)]$$
> > $$=-\frac{2}{d_k(d_k+1)}(\frac{Tr[H^2_l]Tr^2[O]}{d_k^2-1}-\frac{Tr[H^2_l]Tr[O^2]}{d_k(d_k^2-1)}-\frac{Tr[H^2_l]Tr[O^2]}{d_k})$$
> > $$=-\frac{2Tr[H_l^2]}{d_k(d_k+1)}(\frac{Tr^2[O]-d_k Tr[O^2]}{d_k^2-1}),$$
> > where $Tr[H_l^2]=2*\binom{n-2}{k-1}=\frac{2k(n-k)}{n(n-1)}d_k$. We take $Z_0$ as the observable, then $Tr[O]=\frac{d_k(n-2k)}{n}, Tr[O^2]=d_k$ (other observables will also hold with the same magnitude). Thus, we have
> > $$Var_{\theta}[\partial_l C]=-\frac{2}{d_k(d_k+1)}\times\frac{2k(n-k)d_k}{n(n-1)}\times(\frac{\frac{d_k^2(n-2k)^2}{n^2}-d_k^2}{d_k^2-1})$$
> > $$=\frac{4k(n-k)}{(d_k+1)n(n-1)}\times\frac{d_k^2(n^2-(n-2k)^2)}{(d_k^2-1)n^2}$$
> > $$=\frac{4k(n-k)}{(d_k+1)n(n-1)}\times\frac{d_k^2(4nk-4k^2)}{(d_k^2-1)n^2}$$
> > $$=\frac{16k^2(n-k)^2 d_k^2}{(d_k+1)n^3(n-1)(d_k^2-1)}\approx \frac{16 k^2(n-k)^2}{n^4 d_k}.$$ $\blacksquare$
> >
> > We can further analyze that if the $k$ is only 1, then $Var_\theta[\partial_lC]\approx \frac{16}{n^3}$. If the $k=\frac{n}{2}$ on the other hand, $Var_\theta[\partial_lC]\approx \binom{n}{n/2}^{-1}$, which is approximate to exponentially small. This result is consistent with the conjecture that the trainability of the circuit is closely related to $d_k$ and smaller $d_k$ will lead to better trainability. We will make sure a detailed version of the analysis on the trainability will be included in the revision.
> >
> >
> > [1] Cherrat, El Amine, et al. "Quantum Deep Hedging." arXiv preprint arXiv:2303.16585 (2023).

---

> > > ### Author Response · Authors · 2023-11-16
> > >
> > > >***W4: Why did the authors choose the penalty weight in Eq. 16 as $\alpha=[0.5,1, 5,10]$ as people usually set this empirical parameter to a much larger value like 100.***
> > >
> > > **Ans:** Thanks for pointing out this problem, and we agree with you that more results should be included to provide a comprehensive view of the performance. We conduct the experiments on much larger $\alpha$ and the results are shown in the following table. We will fill up these new results in Table 3 in the revision.
> > >
> > >
> > >
> > > |              | BS   | RBS  | $\alpha=10$ | $\alpha=50$ | $\alpha=100$ | $\alpha=1000$ |
> > > | ------------ | ---- | ---- | ----------- | ------------ | ------------ | ------------ |
> > > | Wine Quality | **1.136342** | 1.135580 | 1.099222 | 1.001571 | 0.983495 | 0.997417 |
> > > | Heart Disease | **3.418328** | 3.418183 | 2.963659 | 2.792978 | 2.697520 | 2.717723 |
> > > | Titanic | **1.197799** | 1.197159 | 0.936198 | 0.929956 | 0.916599 | 0.923464 |
> > > | Dry Bean | 8.616844 | **8.616916** | 7.790612 | 7.928807 | 7.864919 | 7.870827 |

---

> > > > ### Author Response · Authors · 2023-11-16
> > > >
> > > > >***Q1: Can you explain why you use gate infidelity as the performance metric for the unitary approximation problem but plot success probability in Fig.2?***
> > > >
> > > > **Ans:** Thanks for your question and we apologize for the unclear explanation for Fig. 2. We use the loss function in equation 12 as the performance metric and set a threshold as $1e-10$. For each unitary matrix, if the loss $\mathcal{L}<1e-10$, we say that we successfully approximate this matrix. The success probability indicates the portion of matrices we can approximate to $\mathcal{L}<1e-10$ from all the 50 randomly generated matrices. We will further explain this in the caption of figure 2.
> > > >
> > > >
> > > > >***Q2: It would be nice if the authors could discuss connections of their HW-preserving VQE with XY-mixer QAOA or other constrained QAOA, and maybe apply the theoretical tools used here to improve those work?***
> > > >
> > > > **Ans:** Thanks for your thoughtful suggestion. The XY-mixer and other mixers are a subset of the proposed HW preserving gates as RBS gates, which we detailly discussed in the literature. The Hamiltonian of the XY-mixer is
> > > > $$
> > > > H_{XY}=\begin{bmatrix}
> > > >         0 & 0 & 0 & 0\\\0 & 0 & 1 & 0 \\\        0 & 1 & 0 & 0\\\0 & 0 & 0 & 0
> > > > \end{bmatrix}
> > > > ,\quad\quad H_{XY}=\begin{bmatrix}
> > > >         0 & 0 & 0 & 0 \\\        0 & a & b & 0 \\\        0 & b^\dagger & c & 0 \\\        0 & 0 & 0 & 0
> > > >     \end{bmatrix}$$ which is a special case of the general HW preserving gates with $b=1$. The unitary matrix of the XY-mixer is $$U_{XY}=\begin{bmatrix}
> > > >         1 & 0 & 0 & 0\\\        0 & \cos (\theta) & -\text{i}\sin (\theta) & 0\\\        0 & -\text{i}\sin (\theta) & \cos (\theta) & 0\\\        0 & 0 & 0 & 1
> > > >     \end{bmatrix}.$$
> > > >     We can utilize the pipeline in this work to analyze the XY-mixer. For the NN connectivity, the dimension of DLA of the XY-mixer is
> > > >     $$dim(g_{XY})=(n+1)(n-1) \qquad n \text{ is odd}$$
> > > >     $$\ \ \quad\qquad \qquad\frac{1}{2}n(n-1) \quad\qquad n \text{ is even.}$$
> > > >
> > > > (This is a piecewise function, but it is hard to write in the markdown. Sorry for the inconvenience!) For the FC connectivity, the dimension of DLA of the XY-mixer is
> > > >
> > > > $$dim(\mathfrak{g}_{XY})= (d_k+1)(d_k-1) \quad n \neq 2k$$
> > > > $$\qquad\quad\qquad\frac{1}{2}(d_k+2)(d_k-2) \quad n = 2k. $$
> > > >
> > > > Thus, we can conclude that XY-mixer with full connectivity is quite capable of solving most of the problems except for some rare cases and maybe that is exactly why it is so famous for solving constrained QAOA. However, the XY-mixer on NN-connectivity is not very good even with the RZZ gate as the phase operator (the dimension of DLA is still not full). Therefore, we can conclude that the XY-mixer might not be so capable considering the physical qubit topology. One possible solution to this is to combine other gates to increase the initial generators and gain full controllability under both connectivities.
> > > >
> > > >
> > > > [1] Hadfield, Stuart, et al. "From the quantum approximate optimization algorithm to a quantum alternating operator ansatz." Algorithms 12.2 (2019): 34.
> > > >
> > > >
> > > > >***Q3: What is the basis set used in ground-state energy estimation to discretize the computation space?***
> > > >
> > > > **Ans:** We do use STO-3G as the computational basis and we will clarify this in the revision, thanks for pointing out this problem.
> > > >
> > > > >***Q4: Feature selection is not a typical benchmark for VQAs. Is this a classically hard problem? Can you say more about why chose this test case?***
> > > >
> > > > **Ans:** Thanks for your comment. We choose feature selection as one of our test cases for the following three reasons:
> > > > 1. Feature selection is a well-studied (2624 citations for [1], 4026 citations for [2], 4926 citations for [3], etc.) problem for the classical machine learning community.
> > > > 2. We already have one VQA problem and we are seeking more diversity by choosing feature selection instead of all the problems listed in [4].
> > > > 3. D-Wave has focused on this problem to demonstrate quantum supremacy on classical machine learning problems with their hardware recently [5]. We believe feature selection is a candidate to fulfill quantum supremacy as a QUBO problem utilizing quantum annealing.
> > > >
> > > > We will further explain these issues in the revision to give the readers a better understanding. Thanks again for your valuable questions and suggestions to help us improve the manuscript.

---

> ### Comment · Reviewer_ADjN · 2023-11-16
>
> For W2: The idea is cool; potentially, you can also add such discussion to the paper.
>
> For W3 and W4, author's answer does address most of my concerns.
>
> For Q1-Q4, authors also give reasonable answers.
>
> Look forward to seeing the revised version of this paper. I decided to raise my score to 8.

---

> > ### Author Response · Authors · 2023-11-17
> >
> > We are truly grateful for your appreciation of our work, and we also thank you for all the thoughtful and valuable suggestions to improve the manuscript! It is really nice to exchange ideas with you.

---

### Author Response · Authors · 2023-11-21
**General Response**

Dear ACs and Reviewers,

We thank all the reviewers for your valuable time and efforts on our paper!

We especially appreciate reviewers for recognizing our paper as novel (ADjN), valuable (Eu68), and with extensive and proper experiments (ADjN, Eu68, and Yeih). We have tried our best to respond to all the reviewers regarding the motivation, trainability, and comparison against previous methods. We have received positive feedback from ADjN and Yeih so far and still working on the discussion with obpr (who raised the contribution score but remained a 3 in the overall rating).

We have uploaded a revised paper based on the suggestions from the reviewers and the modification is marked in blue. Improvements include (1) rewriting the related works to discuss hard-constrained XY-mixer in detail and rearranging the whole section to the appendix, (2) adding the theoretical analysis on the trainability of HW preserving gates, (3) adding a remark on the bit flip error, (4) fixed some typos and unclear description.

Since the discussion session is approaching the deadline (with less than 48 hours), we wonder whether the reviewers' concerns have been resolved by our reply and whether maybe reviewer obpr could consider adjusting the overall rating accordingly.

Best regards,

Authors

---

### Meta-Review · Area_Chair_VNxd · 2023-12-04

**Metareview:**

Variational quantum algorithms constitute a popular approach for algorithms on Noisy Intermediate-Scale Quantum (NISQ) computers, and they share similarities to classical neural network because they are both parametrized and can be optimized to achieve good performance. This paper studies ansatzes for variational quantum algorithms, in particular Hamming Weight (HW) preserving ansatzes that obey symmetries and constraints. Theories are established by quantum optimal control theory and quantum overparameterization theory, and extensive experiments are conducted to verify the superior performance of such ansatzes.

This paper has several strengths:
- Application of Dynamical Lie Algebra (DLA) and over-parameterization theory for parametrized quantum circuits to design symmetry-preserving VQA ansatz.
- Experiments are nicely conducted and support the theoretical claims.

On the other hand, it also has weaknesses:
- The most notable weakness is the novelty - the ansatz is not new, but the analysis and corresponding experiments.
- In terms of theoretical contributions, it is a bit incremental compared to arXiv:2303.16585, especially the lower bound of the gradient and experiments for the constrained VQA problem. There are also a few other papers closely related to the current submission.

There have been detailed discussions during the Reviewer-AC discussion period. The conclusion is that the paper is not perfect, but the rebuttals have made the paper stronger, and given that quantum papers have relatively little presence in ML conferences but the interest is significantly increasing, and the current scores 5, 6, 8, 8 are competitive (the previous score 3 was increased to 5), the final decision by the AC is to mark this paper as borderline accept and will appear in ICLR 2024 as a poster.

For the final version of the paper, the authors should double check that all the points in the rebuttals are included. It would be better if the authors can also explain more beyond the "rethinking" perspective, in particular whether the proposed theoretical analysis and numerical experiments can apply to other ansatzes for variational quantum algorithms.

**Justification For Why Not Higher Score:**

The contribution of this paper is probably not competitive enough for being highlighted as an oral or spotlight.

**Justification For Why Not Lower Score:**

As explained in the meta-review, the paper is not perfect but the rebuttals have made the paper stronger, and given that quantum papers have relatively little presence in ML conferences but the interest is significantly increasing, and the current scores 5, 6, 8, 8 are competitive (the previous score 3 was increased to 5), the final decision by the AC is to mark this paper as borderline accept and will appear in ICLR 2024 as a poster.

---

### Decision · Program_Chairs · 2024-01-16

Accept (poster)